# A genome-wide CRISPR functional survey of the human phagocytosis molecular machinery

Patrick Essletzbichler[1], Vitaly Sedlyarov[1], Fabian Frommelt[1], Didier Soulat[2], Leonhard X Heinz[1], Adrijana Stefanovic[1], Benedikt Neumayer[1], Giulio Superti-Furga[1,3]

Phagocytosis, the process by which cells engulf large particles, plays a vital role in driving tissue clearance and host defense. Its dysregulation is connected to autoimmunity, toxic accumulation of proteins, and increased risks for infections. Despite its importance, we lack full understanding of all molecular components involved in the process. To create a functional map in human cells, we performed a genome-wide CRISPRko FACS screen that identified 716 genes. Mapping those hits to a comprehensive protein–protein interaction network annotated for functional cellular processes allowed retrieval of protein complexes identified multiple times and detection of missing phagocytosis regulators. In addition to known components, such as the Arp2/3 complex, the vacuolar-ATPase-Rag machinery, and the Wave-2 complex, we identified and validated new phagocytosis-relevant functions, including the oligosaccharyltransferase complex (MAGT1/SLC58A1, DDOST, STT3B, and RPN2) and the hypusine pathway (eIF5A, DHPS, and DOHH). Overall, our phagocytosis network comprises elements of cargo uptake, shuffling, and biotransformation through the cell, providing a resource for the identification of potential novel drivers for diseases of the endo-lysosomal system. Our approach of integrating protein–protein interaction offers a broadly applicable way to functionally interpret genome-wide screens.

## Introduction

Phagocytosis is a multifaceted, integrated biological process, that involves the engulfment of large (≥0.5 μm) particles under strict coordination over time and space, with an immunological and homeostatic purpose (Flannagan et al, 2012; Underhill & Goodridge, 2012; Gordon, 2016). It is a key defense and neutralization mechanism against pathogens and at the same time, it is involved in the

clearance of apoptotic cells (Arandjelovic & Ravichandran, 2015), in the processes of wound healing (Giulian et al, 1989), in tumor cell phagocytosis (Feng et al, 2019; Kamber et al, 2021), microglial phagocytosis (Podleśny-Drabiniok et al, 2020), and in the resolution of tissue injury (Gerlach et al, 2021). Among other phenomena, dysregulated phagocytosis or aberrant phagosome maturation can promote Alzheimer's disease, allow pathogens such as bacteria or viruses and cancer cells to escape destruction, or promote autoimmunity through incomplete degradation of cell debris and DNA (Arandjelovic & Ravichandran, 2015; Podleśny-Drabiniok et al, 2020).

The material taken up by a cell is contained in a membrane-bound vacuole called phagosome that matures through a series of highly organized membrane fusion and fission events, altering its composition and gradually acidifying its pH (Flannagan et al, 2012). A freshly formed phagosome (pH ~7.4) fuses first with an early endosome acquiring, among others, the GTPase Rab5. Then it undergoes a series of maturation steps, yielding the late phagosome marked by the exchange of Rab5 to Rab7 and the acquisition of additional vacuolar ATPase proton pumps, which acidify the phagosome to pH 5.5–6.0, forming the ideal degradative environment (Flannagan et al, 2012).

Systematic searches for phagocytosis regulators have been widely undertaken with RNAi-based screens in cultured *Drosophila* cell models (Rämet et al, 2002; Kocks et al, 2005; Philips et al, 2005). Until recently, such genetic screens have been lacking mammalian cell systems (Haney et al, 2018; Sedlyarov et al, 2018; Yeung et al, 2019; Lindner et al, 2021). The advancement of gene-editing tools, in particular the discovery of CRISPR/Cas9 systems (Cong et al, 2013; Jinek et al, 2013; Mali et al, 2013) and increasingly sophisticated read-out systems, now allows conducting large genome-wide knock-out screens in mammalian cells at high precision (Cong et al, 2013; Sanjana et al, 2014; Shalem et al, 2015).

Once it is established that a reporter system faithfully represents the biology under investigation and once validation and calibration of the system allow it to display the necessary signal-to-noise ratio, pooled FACS-based genetic screens offer a powerful method to

[1]CeMM Research Center for Molecular Medicine of the Austrian Academy of Sciences, Vienna, Austria    [2]Institute of Clinical Microbiology, Immunology and Hygiene, Universitätsklinikum Erlangen and Friedrich-Alexander-Universität Erlangen-Nürnberg, Erlangen, Germany    [3]Center for Physiology and Pharmacology, Medical University of Vienna, Vienna, Austria

Correspondence: gsuperti@cemm.oeaw.ac.at
Vitaly Sedlyarov's present address is Boehringer Ingelheim RCV GmbH & Co KG, Vienna, Austria
Leonhard X Heinz's present address is Division of Rheumatology, Department of Internal Medicine III, Medical University of Vienna, Vienna, Austria

genetically map a large variety of biological processes (Doench, 2018; Bock et al, 2022).

Here, we present a FACS-based genome-wide CRISPR/Cas9 knock-out screen for phagocytosis using a reporter assay (Colas et al, 2014) that we previously employed for focused genetic screening (Sedlyarov et al, 2018) and for measuring the maturation of the phagolysosome (Heinz et al, 2020). As in vitro macrophage cell model, we chose PMA-differentiated human THP-1 cells, a classical cell model, to study phagocytosis and immune modulation (Fleit & Kobasiuk, 1991; Chanput et al, 2014).

The study reported here represents a comprehensive genetic assessment of cellular functionalities involved in phagocytosis, including its regulation, dynamic phagosomal maturation processes, and movement by the complex cell cargo transport machinery. Additionally, the study highlights the power of combining the information from functional genetic screening and publicly available protein–protein interaction (PPI) data, representing the physical base for concerted cellular activities. We present the efficient guiding of validation and rationalization of the genetic results in light of the annotated cellular machinery and its individual components.

# Results

## A FACS reporter-based screen for modulators of phagocytosis

Phagocytosis proceeds through defined key steps, including the uptake of cargo, the systematic coordinated transport to the lysosome, and proper acidification. We set out to identify genes potentially involved in the entire process by making use of pooled CRISPR/Cas9 knock-out screening in combination with reporter-based flow cytometry-assisted cell sorting. For this, we chose the myeloid cell line THP-1 (Chanput et al, 2014) and a PMA-based differentiation protocol that induces in these cells a macrophage-like phenotype (Fleit & Kobasiuk, 1991). To validate the experimental setup, we tested actin polymerization using Cytochalasin D (Carter, 1967) and lysosome-dependent acidification using Bafilomycin A1 (Fig 1B and C) (Dröse et al, 1993). Next, we introduced a genome-wide CRISPR/Cas9 knock-out library, targeting all protein-coding human genes with 70,948 sgRNAs and 142 control sgRNAs (Hart et al, 2017). We used the lentiCRISPRv2 single vector driving the expression of a sgRNA-cassette and Cas9 developed by Sanjana and colleagues (Sanjana et al, 2014).

To measure the phagocytic uptake of cargo as well as the transport and acidification of the resulting phagolysosome, we employed a flow cytometry readout based on opsonized 1.75-$\mu$m latex beads coated with the pH-sensitive dye pHrodo (Colas et al, 2014) and successfully adopted for genetic screening (Sedlyarov et al, 2018). Whereas cells that were fully functional in phagocytosis and acidification appeared in the PhagoLate fraction, cells still capable of phagocytosis, but deficient in acidification, emerged in the PhagoEarly fraction (Fig 1A). In contrast, cells that were completely deficient in phagocytosis appeared in the PhagoNeg fraction. Since the cytoskeleton has a key role in the formation of the phagocytic cup, and for material uptake, we employed the cytoskeletal

inhibitor Cytochalasin D to show the efficient blockage of reporter uptake. Cytochalasin D–treated cells emerged predominantly in the PhagoNeg fraction, indicating that their phagocytic defect stemmed from an uptake issue (Fig 1B and C). We treated cells with a vacuolar-ATPase inhibitor Bafilomycin A1 to show that acidification of the lysosome was required for cells to emerge in the PhagoLate gate. The uptake of the reporter itself remained unaffected, which led to a rise of cells in the PhagoEarly gate (Fig 1B and C).

## A genome-wide screen identifies known and novel modulators of phagocytosis

After setting up the phenotypic assay to measure phagocytosis in THP-1 cells, we next performed a genome-wide screen for modulators of phagocytosis in quadruplicates following the workflow portrayed in Figs 1A and S1. We first introduced a genome-wide CRISPR knock-out library into THP-1 cells and selected cells with puromycin before we induced a macrophage-like phenotype via PMA differentiation. The period chosen for further cell incubation was the time point where further incubation did not lead to additional phagocytosis (3 h), whereas the bead-to-cell ratio was optimized to yield ~one-third of cells in the PhagoNeg fraction and two-thirds of the cells in the phagocytosis- and acidification-positive fractions (PhagoEarly and PhagoLate) (Fig S2A). The stability of the assay was confirmed by storing cells for 24 h at 4°C which led to no significant change in the reported phagocytosis rate, compared to our standard 3-h condition (Fig S2B). Cell fractions were gated and FACS sorted and processed separately allowing for comparison of sgRNA abundance within each population and replicate (Fig S3A and B).

By comparing sgRNAs that were depleted in PhagoLate compared to PhagoNeg, we identified in total 716 genes to be involved either in phagocytosis itself or in phagosome acidification (Fig 1D and Table S1). Gene Ontology (GO) enrichment analysis for "Cellular components" GO terms yielded among others as strongly enriched compartments the lytic vacuole membrane, lysosome, and late endosome (adj. $P$-value < $1.08 \times 10^{-6}$). These components are all part of cellular processes and pathways which are well-known to be involved in phagocytosis (Fig 1E). Enriched biological processes included the dolichol-linked oligosaccharide biosynthetic process, TOR signaling, starvation response, and actin nucleation (adjusted $P$-value < $5.30 \times 10^{-6}$) (Fig 1E). RNA-sequencing of THP-1 cells treated with the reporter was further used to confirm that genes identified as phagocytic regulators were expressed in THP-1 cells across a wide range of expressions (Fig S4A and B and Table S3). By comparing sgRNAs that were depleted in cells right before differentiation (day 21 after infection) with the CRISPR library plasmid stock, we identified genes that led to a general reduction in growth in our screen (Table S2). We highlighted those genes as "essential genes" in Fig 1D and showed that this set of hits compares well with genes that have a gene effect score <–1 in the DepMap CRISPR dataset (DepMap, 2022) (Fig S5A). In addition, our essential genes show large overlaps (723 of 1041) with the "core essentialome," a list of 1,736 shared essential genes we have previously derived from HAP1 and KBM7 cell lines (Blomen et al, 2015) (Fig S5B). Although our screen could still identify several phagocytosis regulators that were essential, the majority of the 716 genes depleted in PhagoLate versus PhagoNeg did not have a gene effect score <–1 in

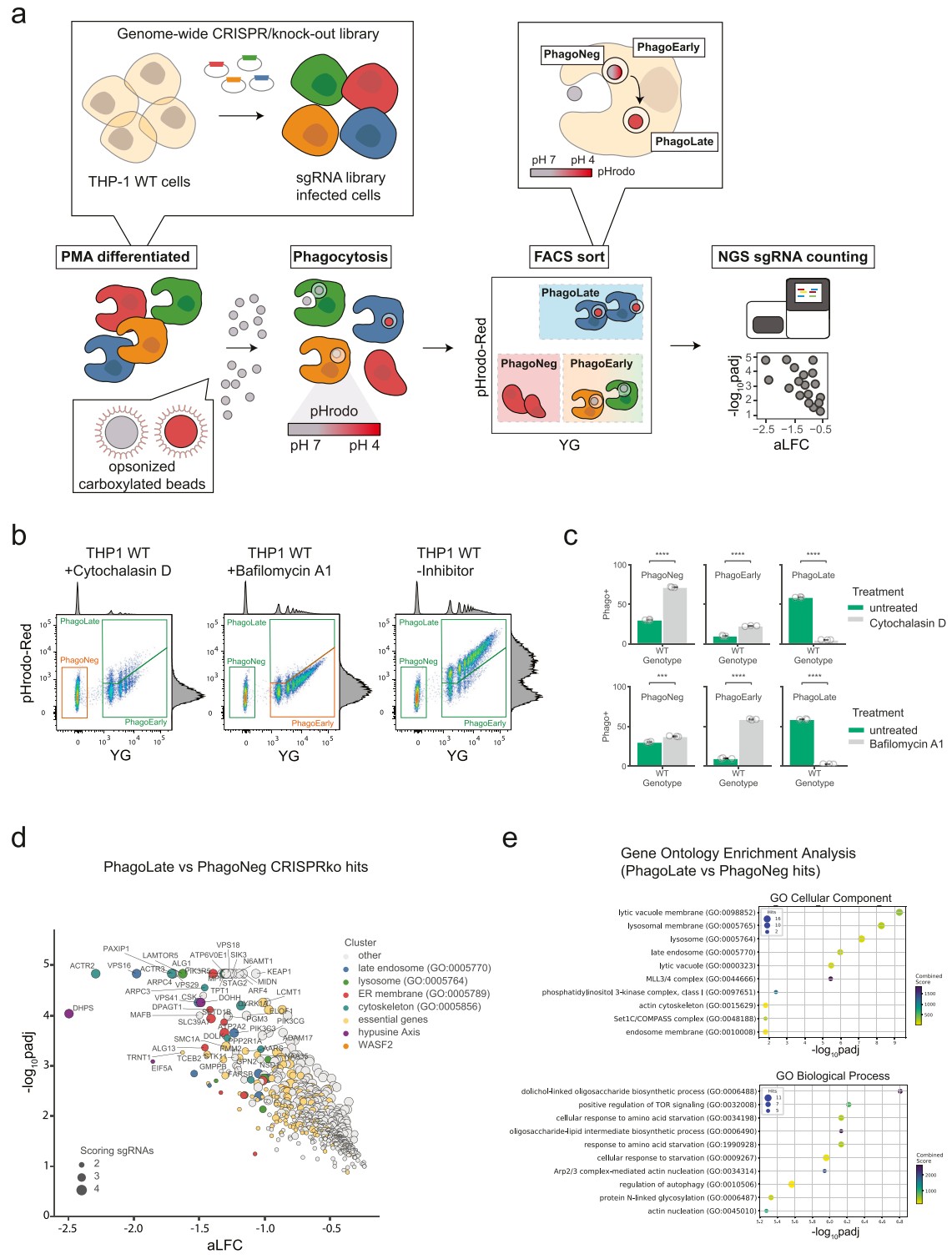

**Figure 1. Genome-wide CRISPR/knock-out approach identifies regulators of phagocytosis.**

**(A)** Schematic overview of the FACS-based genome-wide CRISPR/Cas9 knock-out screen to identify modulators of phagocytosis. **(B)** Representative flow cytometry scatterplots of phagocytosis assays. PMA-differentiated THP-1 cells were either treated with the control drugs Cytochalasin D or Bafilomycin A1 or were left untreated and then incubated with opsonized, dual-colored reporter beads. Each dot corresponds to one measured cell: The intensity of the pH-insensitive dye is represented on the x-axis (YG, striped pattern originates from the number of beads within the individual cell), and the signal of the pH-sensitive dye, which increases signal with decreased pH is displayed on the y-axis (pHrodo-Red). Double-negative cells were sorted as phagocytosis-negative (PhagoNeg), single-positive cells (YG high signal but low pHrodo-Red signal) were grouped as early stages of phagocytosis and double-positive cells (YG high signal and high pHrodo-Red signal) were classified as cells that underwent phagocytosis and phagosome acidification (PhagoLate). Cells treated with the control drug Cytochalasin D remained mostly in the PhagoNeg gate and cells treated with

DepMap (Fig S5C). For all the downstream analyses, we, therefore, worked with the complete set of 716 genes.

## Mapping the identified phagocytosis genes to the cellular machinery

The virtually compiled assembly of human cellular protein complexes inferred from collective PPI studies is sufficiently advanced to represent a useful framework for modeling the molecular machinery of a cell. If our functionally determined genes were indeed representing important elements in the machinery involved in phagocytosis, they should converge on a limited number of protein complexes. Moreover, these protein complexes should in turn be involved in linked cellular processes and pathways. To test this hypothesis and to probe for physical interactions between the 716 genetic hits, we created a PPI network, derived from interactions recorded in the BioPlex 3.0, a high-quality dataset featured by the consistency of scoring across different baits (Huttlin et al, 2021). For 69% of the 716 hits, we could obtain a fairly coherent network with only a few small, isolated subnetworks. For a better interpretation of the network, the color of the circle represents the average log$_2$-fold change (aLFC) of each gene (Fig 2). To map the proteins linked in the protein network to known molecular machines, we scored the membership to the well-characterized human protein complexes of the CORUM-core (Giurgiu et al, 2019), light brown shaded in Fig 2 (see also Table S4). This allowed us to map about a third of the identified phagocytosis modulators onto known biological complexes representing molecular machines (Fig 2). Additionally, the completeness of protein-complexes annotated in the CORUM-core was analyzed and indicated in percentages with either a blue (percent of the complex covered by all the genetic hits scoring in the screen) or a yellow pie chart (percent of the complex covered by all the genetic hits which have interactions reported in the Bioplex 3.0 dataset), whereas the diameter of the circle indicates the total size of the reported complex (Table S5).

## Actin cytoskeleton nucleation plays an important role in phagocytosis

To ratify our dataset of phagocytic modulators, we decided to pick genes for an arrayed validation that displayed the largest aLFC from the identified complexes with the highest completeness, underlaid with light orange ellipses in the network (Fig 2). Among these was the Arp2/3 complex (Goley & Welch, 2006). This actin-nucleating machinery has been described to be highly essential for phagocytic cup formation and is also the mechanistic target of our chemical assay control, Cytochalasin D (May et al, 2000). Our genetic screen picked up all subunits of the Arp2/3 complex (ARPC1B, ARPC2, ARPC3, ARPC5, ARPC4, ACTR3, and ACTR2), highlighting the high

degree of coverage of our experimental set-up. Assessing the reporter uptake of individual *ARPC2* KO cell lines showed significantly decreased phagocytosis (Fig 3A and B), validating the findings of the genome-wide pooled screen. Moreover, our screen identified 80% of the members of the Wave-2 complex as hits (WASF2, CYFIP1, ABI1, BRK1, and NCKAP1) and most of these had recorded interactors in the BioPlex 3.0 dataset. WASF2 is a scaffolding protein that employs the Arp2/3 complex and has been reported to be essential for the formation of lamellipodia (Oikawa et al, 2004), indicating that our reporter enters macrophages through these structures. Since WASF2 is a central actin-nucleator and scaffolding protein for Arp2/3, without being essential for diverse processes within the cell unrelated to phagocytosis as the Arp2/3 complex, we picked it as the first in-detail validation target (Fig 3C). Thus, we derived *WASF2* KO clones from the human monocyte cell line U937 (Liu & Wu, 1992), which is competent for phagocytosis and amenable to single clone isolation. We used 4 different sgRNAs targeting different exons of *WASF2* (Fig 3G). After confirming by immunoblotting, the successful knock-out of *WASF2* in these clones (Fig 3D), we rescued them either with a *WASF2* cDNA (designed to be resistant against sgRNAs 1, 2, and 4 but not against sgRNA 3, cutting control that can cut the *WASF2* cDNA, Fig 3G) or with mock cDNA as control. WASF2 is proposed to be a key effector in the life cycle of *Listeria monocytogenes* (Bierne et al, 2005). We took advantage of this to validate our cellular clones. We infected them with *Listeria* and quantified the total amount of bacteria phagocytosed after 3 h, as exemplified in Fig 3E. Effectively, the *WASF2* knock-out cell lines that only express the mock control cDNA, harbored lower levels of total phagocytosed bacteria (Fig 3F). The phagocytic activity of those cells was then evaluated using the reporter uptake assay. Indeed, cells knocked out for *WASF2*, only expressing the mock control cDNA, showed significant ablation for the PhagoLate fraction with a concurrent increase of the PhagoNeg fraction, indicating a decrease in total phagocytosis (Fig 3H). At the same time, this defect could be rescued by expressing a *WASF2* cDNA, confirming that the impairment of phagocytosis was caused by the knock-out of *WASF2* (Fig 3G and H).

Our physical interaction map of genetic hits offers the opportunity to streamline validation by navigating neighboring cellular complexes as a functional framework, instead of choosing randomly among the 716 genes (Fig 2). The underlying assumption is that proximity to a validated node will increase the validation efficiency of the next one, while allowing to derive a functional relationship between the two.

In our network, the Wave-2 complex is connected via BRK1 to the WASH complex (KIAA0196, CCDC53), an endosomal hub responsible for the activation of Arp2/3 (Jia et al, 2010) (Fig 2). Knock-out of *KIAA0196* indeed led to a phenotype (significant reduction of reporter uptake) comparable to *WASF2*, confirming the functional connection suggested by the protein network (Figs 2 and 3A).

Bafilomycin A1 stayed in the PhagoEarly gate. The marginal intensity distributions are illustrated on the side of the plot. **(B, C)** Barplot showing the quantification of the different fractions of cells gated in (B). Data are mean ± 95% confidence interval from 3 technical replicates, representative for two independent experiments. ***$P \le 0.001$, ****$P \le 0.0001$; by Welch's $t$ test. **(D)** Volcano plot showing the 716 genes depleted in the PhagoLate population versus the PhagoNeg population. The x-axis shows the average log$_2$-fold change calculated for all sgRNAs per gene against the y-axis, representing the statistical significance as −log$_{10}$Padj. The size of the dot indicates the amount of sgRNAs changing significantly for the particular gene. **(D, E)** Gene Ontology (GO) enrichment analysis (two-sided Fisher's exact test, $P$-value adjusted for multiple testing) for genes depleted in the PhagoLate population seen in (D) for GO cellular components and GO biological processes. The x-axis shows the significance of enrichment (−log$_{10}$-transformed $P$-value adjusted for multiple testing) against the y-axis showing the top 10 enriched terms, sorted by adjusted $P$-value.

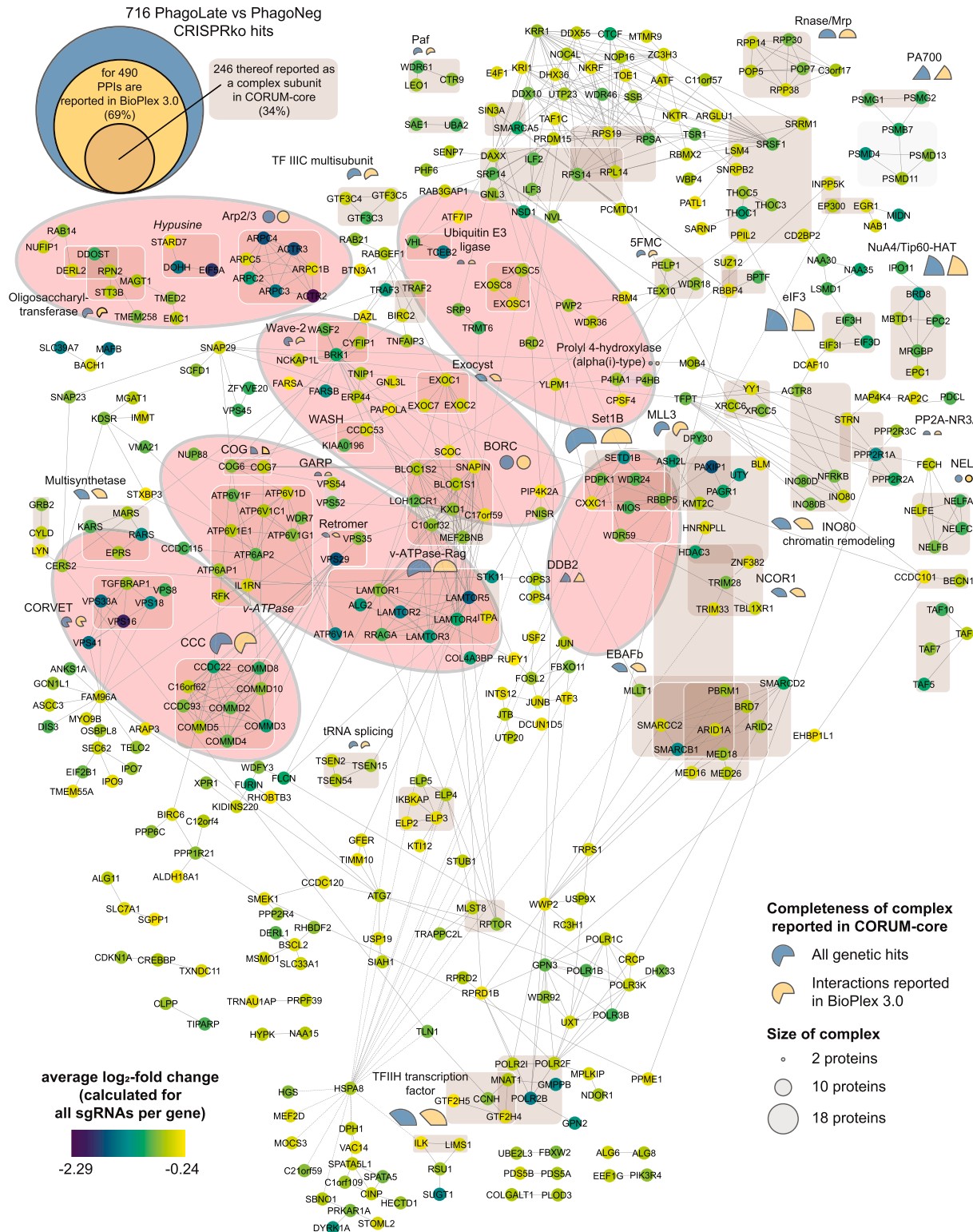

**Figure 2. Physical interaction map of genetic hits yields the molecular network regulating phagocytosis.**

Network of 490 genes that are depleted in the PhagoLate population (Fig 1D) and have reported protein–protein interactions in BioPlex 3.0. Each circle represents a gene, and the lines represent the reported protein–protein interactions. The color of the circle represents the average log$_2$-fold change. 246 of the genes are reported as a complex subunit in the CORUM-core database and are highlighted with light brown boxes. The percent completeness of these complexes is indicated either with a blue (percent of respective complex covered by all the genetic hits scoring in the screen) or a yellow (percent of respective complex covered by all the genetic hits which

## Lysosomal regulation modulates acidification of the lysosome and affects phagocytosis

Continuing along our "validation path" we observed that the WASH complex is connected multiple times to the BORC complex (BLOC1S1, BLOC1S2, SNAPIN, LOH12CR1, KXD1, C10orf32, C17orf59, and MEF2BNB) (Pu et al, 2015), which we identified in our genetic screen with a remarkable 100% completeness, underlining its importance for phagocytosis. Since the BORC complex in our network displays further connections to the vacuolar–ATPase–Rag complex (LAMTOR1, LAMTOR2, LAMTOR3, LAMTOR4, LAMTOR5, ALG2, ATP6V1A, and STK11) and is through these further connected to the vacuolar-ATPase protein machinery (ATP6V1F, ATP6V1D, ATP6V1C1, ATP6V1E1, ATP6V1G1, ATP6AP1, ATP6AP2, WDR7, and IL1RN) (Kissing et al, 2015), we believe that this subnetwork of hits delineates important lysosome-regulating functions. ARL8B is another prominent regulator of lysosomal trafficking that scored in our screen and is part of this subnetwork, but is not represented in our network as it lacks within BioPlex 3.0 any direct connection with hits identified within our screen (Garg et al, 2011; Pu et al, 2015). We validated this neighborhood by knocking out *LAMTOR5* as a member of the vacuolar–ATPase–Rag complex and *KXD1*, a member of the BORC complex. Whereas *LAMTOR5* loss of function cell lines showed a significant reduction in phagocytosis, knock-out of *KXD1* led to less or no significant effects in the individual phagocytosis assays. This result correlates with the obtained aLFC for *LAMTOR5* (aLFC = −1.62, −$\log_{10}P$adj = 4.83) versus *KXD1* (aLFC = −0.696, −$\log_{10}P$adj = 2.29) in the genetic screen. We believe that this case demonstrates that single-cell line validation experiments often struggle to significantly validate small effects on a biological process, whereas on the other hand, pooled genetic screening that uses hundreds of millions of individual cells has the experimental and statistical power to detect the entire BORC complex as a regulator for phagocytosis (Fig 3A). Interestingly, the target of Bafilomycin A1, vacuolar-ATPase, is also a hit in this subnetwork. In contrast to long-lasting genetic ablation, as shown with *ATP6AP2* knock-out cell lines (Fig 3A), acute drug treatment allows revealing the importance of the acidification process of phagocytosis without affecting uptake or cargo shuffling (Fig 1B).

### Involvement of the vesicle-trafficking machinery in phagocytosis

In our network of functional hits, the BORC complex was also connected to the conserved structural Golgi protein COG6 (Ungar et al, 2002) and through this, to four complexes responsible for vesicle trafficking within the cell, such as the exocyst complex (EXOC1, EXOC2, and EXOC7) (Katoh et al, 2015), the GARP complex (VPS52 and VPS54) (Pérez-Victoria et al, 2010), the retromer complex (VPS29 and VPS35) (Haft et al, 2000), and the CORVET complex (VPS16, VPS18, VPS33A, VPS8, and TGFBRAP1) (van der Kant et al, 2015). To continue our validation campaign, we generated knock-out cell lines for several subunits of the abovementioned complexes including *COG6*, *EXOC1*, *VPS52*, *VPS35*, and *VPS16*. We observed a

significant reduction in phagocytosis in all these knock-out cell lines (Fig 3A).

The knock-out of *VPS35*, a subunit of the retromer complex, significantly reduced the PhagoNeg and interestingly increased the PhagoEarly fraction. This indicates that in a functional assessment of the different phagocytosis steps, absence of *VPS35* increases the uptake of the reporter, while leaving the rate of acidification of material unchanged. This could represent the equivalent of a "traffic jam" in the early endosome (PhagoEarly). Knock-out of *COG6* seems to affect the uptake of beads less, but rather the transport to the phagolysosome (Fig 3A).

In our network, we could then observe interactions between the retromer complex and all of the subunits of the CCC complex (also known as the Commander Complex). Within our genetic screen, we covered all 10 in CORUM-reported subunits of the CCC complex (COMMD2, COMMD3, COMMD4, COMMD5, COMMD8, COMMD10, CCDC22, CCDC93, VPS29, and C16orf62/VPS35L) as single individual hits. The CCC complex is a poorly studied protein assembly that seems to play a role in endosomal cargo retrieval, recycling, and regulation of NF-κB and hypoxia-induced transcription (Cullen & Steinberg, 2018; Laulumaa & Varjosalo, 2021). Through the strong interconnection with the retromer complex within the network presented in this work (Fig 2), we speculate to ascribe a biological role in phagocytosis to the CCC/Commander complex, likely to be related to the vesicular sorting function of the retromer complex (Figs 2 and 3A). Assessing uptake of the reporter in individual *COMMD3* KO cell lines showed variations between replicates of the phagocytosis assay. This may reflect a potentially more pleiotropic effect of the CCC complex on phagocytosis (Fig 3A), as one might expect based on the above-reported features of the CCC complex.

### The oligosaccharyltransferase complex (OST) is a strong modulator of phagocytosis

Although the OST (RPN2, DDOST, STT3B, and MAGT1) forms an independent subnetwork on our map, we decided to further assess its role due to its high completeness in our screen. The OST complex is a core-member of the central machinery for N-linked protein glycosylation. We chose to knock out the magnesium transporter *MAGT1/SLC58A1*, known to be mutated in patients with a disorder called X-linked immunodeficiency with magnesium defect, EBV infection, and neoplasia (XMEN) (Ravell et al, 2020). A further hint that SLC58A1 is involved in phagocytosis is that *SLC58A1*-deficient cells manifested a very strong defect in phagocytosis (Fig 3A).

We chose to validate six additional hits from the screen that were not part of large molecular complexes but had not been previously associated with phagocytosis. Of these hits, three showed a less robust effect in the validation process, manifested by variations between individual replicates of phagocytosis assays (*COPS3*, *P4HA1*, and *PSMD13*). Knock-out of *STK11* mainly negatively affected the uptake of the material, revealed by decreased PhagoNeg fractions but fewer effects on the PhagoEarly and PhagoLate

---

have interactions reported in the Bioplex 3.0 dataset) pie chart, whereas the size of the circle indicates the total size of the reported complex. Identified complexes with the highest completeness are underlaid with light orange ellipses.

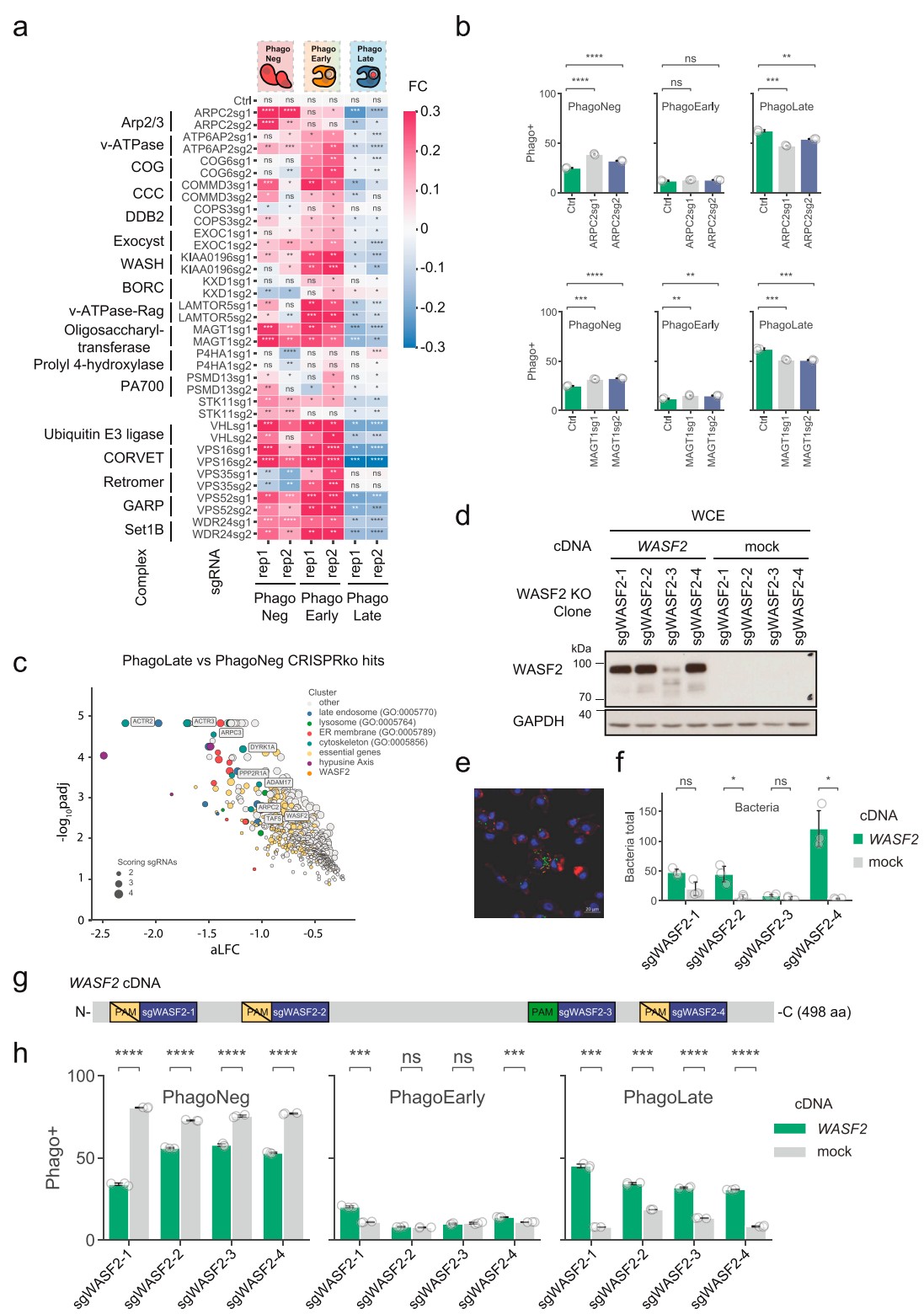

**Figure 3. Systematic validation of phagocytosis hits.**
**(A, B)** Heatmap showing the fold change (FC) of fractions recorded in phagocytosis assays in the respective knock-out cell lines listed on the y-axis compared to control cell lines (Ctrl), as shown in an example in (B). Blue color indicates negative FC, white color no FC, and red color positive FC. For each sgRNA phagocytosis assays were performed in two biological replicates (indicated as rep1 and rep2 on the x-axis), consisting of three technical replicates each, as indicated on the x-axis. Each cell represents the mean ± 95% confidence interval from three technical replicates. ns $P > 0.05$, *$P ≤ 0.05$, **$P ≤ 0.01$, ***$P ≤ 0.001$, and ****$P ≤ 0.0001$; by Welch's $t$ test.
**(B)** Barplot showing the quantification of the PhagoNeg, PhagoEarly, and PhagoLate fractions as examples for validation of the Arp2/3 complex on *ARPC2* KO cell lines and

fractions and Ubiquitin E3 ligase *VHL*, and *WDR24*, led to a general defect in phagocytosis (Fig 3A).

In summary, of the genes that we assessed individually in phagocytosis assays tracking three different stages, more than three quarters validated effectively. This suggests that overall, the dataset encompassing 716 genes represents a valuable resource for the molecular characterization of the cellular machinery in the complex phagocytosis process.

### Depletion of the unique post-translational modification hypusine on eIF5A reduces phagocytosis

Interestingly, the gene with the highest significant depletion in genome-wide phagocytosis screen was Deoxyhypusinesynthase (*DHPS*), scoring together with two other genes (*DOHH* and *eIF5A*) that are all involved in forming the unique post-translational modification hypusine (Park et al, 1981) (Figs 1D and 4G). eIF5A is a small ~17-kD conserved protein, the mammalian ortholog of the bacterial translation factor EFP (Doerfel et al, 2013), and the only known target of the post-translational modification hypusine, which is incorporated by an interplay of the two enzymes DHPS and DOHH from the polyamine precursor—spermidine (Wolff et al, 2007). Since not only *eIF5A* but also *DHPS* and *DOHH* scored as hits in the screen, we concluded that the absence of hypusine must be causative of a phagocytosis defect (Fig 4A). To check this, we generated an inducible THP-1 knock-out model for *DHPS* using an inducible CRISPR-Cas9 knockout backbone (TLCV2) (Barger et al, 2019). Furthermore, since there was no commercial antibody available for hypusine, we generated a recombinant hypusine-specific antibody based on a sequence published by Zhai and colleagues (Zhai et al, 2016). We validated the antibody by transiently expressing *DHPS*, *DOHH*, and *eIF5A* (either *eIF5A*-WT or mutated for the hypusine-modification site lysine 50) in HEK293T cells. The antibody successfully detected hypusinated eIF5A, whereas there was no signal detected in cells overexpressing the eIF5A-K50 variant (Fig 4B). To further validate the binder, we tested it on lysates of THP-1 cells treated with the DHPS-inhibitor GC7 (Melis et al, 2017) and observed down-regulation of hypusine after 24–72 h (Fig 4C). As GC7 could also act indirectly, we used the inducible *DHPS* knock-out cell models to show that the down-regulation of DHPS is sufficient to down-regulate hypusine (Fig 4F). Having validated our experimental set-up, we measured phago-cytosis in both settings. Treating THP-1 cells with GC7 led to a

significant drop of the PhagoLate fraction with a parallel rise of the PhagoNeg fraction, suggesting an effect of hypusine on the total rate of phagocytosis (Fig 4D). We observed the same effect when we measured phagocytosis after *DHPS* knock-out (Fig 4H). Introducing a *DHPS* WT cDNA, designed to be resistant against the sgRNA, could rescue the defect, demonstrating the specific dependency on *DHPS* (Fig 4E, F, and H). As an additional control of our cell model, we then rescued our *DHPS* knock-out model with *DHPS* cDNAs mutated to strongly reduce spermidine binding (D243A) (Lee et al, 2001). In-terestingly, when we rescued our cell model with the D243A mutant *DHPS* cDNA, the cell seemed to compensate for the reduced function of the D243A *DHPS* variant by higher expression levels of D243A compared to other DHPS protein levels (Fig 4F). Nevertheless, despite higher expressed levels of the mutant DHPS, we observed less phagocytosis compared to cells rescued with *DHPS* WT cDNA (Fig 4H). A clinical study on patients with severe neurodevelopmental disorders reported recessive rare variants in the *DHPS* gene (Ganapathi et al, 2019). Therefore, we wondered if the expression of these mutant DHPS variants would also lead to a defect in our phagocytosis assay. Indeed, when we expressed either the N173S or the ΔY305–I306 patient *DHPS* variants in our *DHPS* knock-out model, we could observe a clear defect in phagocytosis (Fig 4H). The expression levels of the mutant DHPS proteins were comparable to the WT (Fig 4F). Thus, *DHPS*, *DOHH*, and *eIF5A* can be regarded as genes validated for their involvement in phagocytosis.

## Discussion

The FACS-based genome-wide CRISPR/Cas9 knock-out screen allowed us to identify a large number of genes involved in the regulation and execution of the phagocytosis process. The high number of genes scoring positive in the assay represented a challenge for further validation. We designed a strategy that used integration with existing PPI networks to prioritize which genes to validate. We reasoned that focusing on those molecular machines of the cell, whose components were "genetically hit" multiple times in the screen, would be an expedient to increase the statistical significance of the various cellular functions associated with phagocytosis and thus ensure a higher validation rate. Recently, Haney and colleagues undertook a systematic search for phagocytic regulators with a pooled CRISPR screen approach (Haney et al, 2018). They used macrophage-differentiated U937 cells and incubated them

---

for the oligosaccharyltransferase complex on *MAGT1* KO cell lines. Data are mean ± 95% confidence interval from three technical replicates, representative for two independent experiments. ns $P > 0.05$, $**P ≤ 0.01$, $***P ≤ 0.001$, and $****P ≤ 0.0001$; by Welch's *t* test. **(C)** Volcano plot highlighting *WASF2* and genes of the Arp2/3 complex scoring within the 716 genes depleted in the PhagoLate population. The x-axis shows the average $\log_2$-fold change calculated for all sgRNAs per gene against the y-axis, representing the statistical significance as $-\log_{10}(P\mathrm{adj})$. The size of the dot indicates the amount of sgRNAs changing significantly for the particular gene. **(D, G)** Western blot showing the expression of WASF2 in U937 *WASF2* knock-out cell clones, overexpressing either a mock cDNA or a *WASF2* cDNA, designed to be resistant to cleavage by sgWASF2-1, 2, and 4 but not sgWASF2-3, as depicted in (G). **(E)** Representative image showing the readout used for quantification of the listeria assay. The green color highlights *Listeria monocytogenes*, the red color highlights actin, and blue highlights the DAPI staining. **(D, F)** Barplot showing the quantification of the total amount of *Listeria monocytogenes* of cell lines validated in (D). The green color highlights the clonal U937 *WASF2* KO cell lines expressing WASF2 and the grey color shows the same cell lines expressing a mock cDNA as control. Data are mean ± 95% confidence interval from three technical replicates. ns > 0.05, $*P ≤ 0.05$ by Welch's *t* test. **(D, G)** Schematic representation of the *WASF2* cDNA overexpressed in cell lines validated in (D). PAM indicates the protospacer-adjacent motif compatible with Cas9 detection of each *WASF2* sgRNA, indicated by the blue boxes. **(D, H)** Barplot showing the quantification of the PhagoNeg, PhagoEarly, and PhagoLate fraction of cell lines validated in (D). The green color highlights the clonal U937 *WASF2* KO cell lines expressing WASF2, and the grey color represents the same cell lines expressing a mock cDNA as control. Data are mean ± 95% confidence interval from three technical replicates, representative for two independent experiments. ns $P > 0.05$, $***P ≤ 0.001$, and $****P ≤ 0.0001$; by Welch's *t* test.

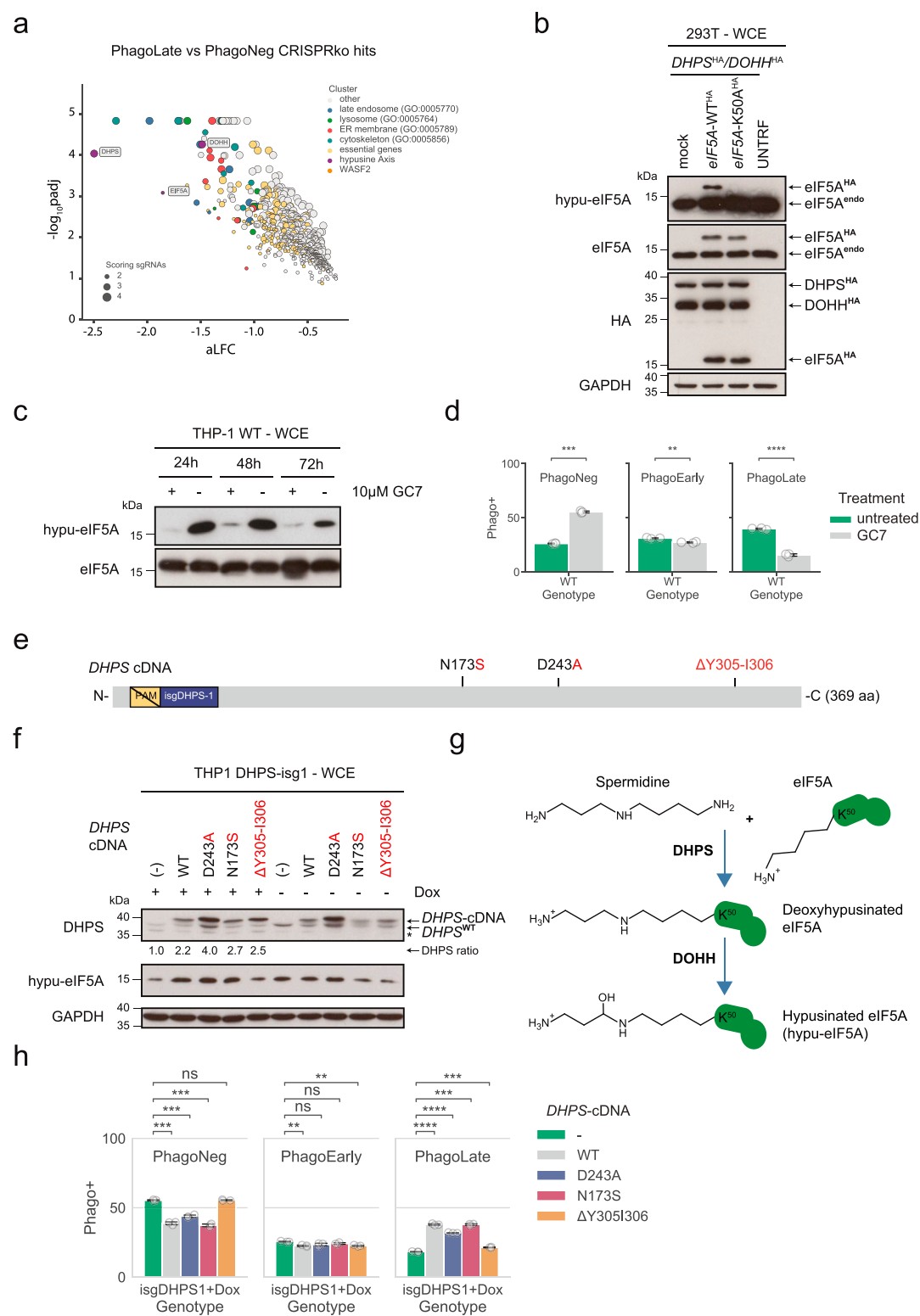

**Figure 4. Absence of hypusine abrogates phagocytosis.**
**(A)** Volcano plot highlighting the genes of the hypusine axis within the 716 genes depleted in the PhagoLate population. The x-axis shows the average log$_2$-fold change calculated for all sgRNAs per gene against the y-axis, representing the statistical significance as −log$_{10}$(Padj). The size of the dot indicates the amount of sgRNAs changing significantly for the particular gene. **(B, C)** Western blot analysis of hypusine (hypu-eIF5A) in HEK293T cells overexpressing either a mock or C-terminal HA-tagged eIF5A-WT (eIF5A-WT$^{HA}$) or eIF5A-K50A mutant cDNA (eIF5A-K50A$^{HA}$) and a C-terminal HA-tagged DHPS cDNA (DHPS$^{HA}$) and DOHH cDNA (DOHH$^{HA}$) (C). Western blot analysis of hypusine levels (hpu-eIF5A) in THP-1 WT cells treated with or without 10 µM GC7 at the respective time point (24–72 h). **(D)** Barplot showing the quantification of the

with magnetic beads conjugated to a series of different substrates such as myelin, zymosan, and sheep red blood cells. In our study, we use the human monocytic cell line THP-1 differentiated into macrophages and an FACS-sorting-based separation approach. We confirmed and expanded their identified regulators as to the importance of the Arp2/3 machinery, Wave-2 complex, and the mTOR-associated Ragulator complex. This was demonstrated by the high genetic coverage of these protein complexes in our dataset. As already shown by Haney and team in U937 cells, loss of subunits of the actin polymerization machinery is detrimental to phagocytosis in THP-1 cells. Several datasets collected in screening efforts for host regulators in the context of *Salmonella* (Yeung et al, 2019), *Legionella pneumophila* (Jeng et al, 2019), and *Staphylococcus aureus* (Lindner et al, 2021) pathogeneses confirmed the importance of the actin polymerization machinery. This indicates that our pathogen-agnostic reporter approach used in this screen successfully mapped the cellular routes taken by pathogens. Furthermore, we identified several members of the exocyst complex to be required for phagocytosis. Previous studies on the interaction of Cdc42 with the exocyst complex, which was found to be essential for promoting phagocytosis (Mohammadi & Isberg, 2013), helped us to interpret the scoring of the exocyst complex in our screen. As Cdc42 is a Rho GTPase central for actin dynamic (Bonfim-Melo et al, 2018), we can now position the exocyst complex close to the family of actin regulators in our hit-reconstructed protein-interaction network (Fig 2).

Our study also confirmed the importance of the positive regulation of mTOR signaling for phagocytosis (Haney et al, 2018), a finding that was also revealed by the identification of RagA (*RRAGA*) as a regulator of microglial phagocytic flux in a zebrafish screen (Shen et al, 2016). Remarkably, our screen identified all five members of the Ragulator complex (LAMTOR1, LAMTOR2, LAMTOR3, LAMTOR4, and LAMTOR5) as well as *RRAGA*. In addition, we found the full BORC complex, the machinery responsible for positioning lysosomes. The BORC complex is reported to interact with the Ragulator, a complex that negatively regulates the activity of BORC in response to amino acid starvation sensed by SLC38A9, a critical regulator of the lysosomal function that we had previously identified (Rebsamen et al, 2015; Filipek et al, 2017; Wyant et al, 2017). The role of BORC in phagocytosis can potentially be explained by a recent preprint showing the importance of TORC1, BORC, ARL-8, and kinesin-1 for vesiculation of the phagolysosome and especially the role of BORC as a promoter for phagolysosomal degradation (Fazeli et al, 2022 *Preprint*).

Our screen also highlighted the importance of coordinated cargo transport through the cell for phagocytosis. This requires the trafficking and fusion of endosomes, controlled by Rab-GTPases, SNARE proteins, and multi-subunit–tethering complexes (Bröcker et al, 2010). Using our dataset to identify the parts of this canonical vesicle-trafficking machinery of most relevance for phagocytosis, we observed that large parts of the CORVET and HOPS complexes, machinery that coordinate together with Rab5 and Rab7 (*RAB5C* and *RAB7A* are hits as well) the fusion of the endosome with the lysosome, score as strong phagocytic regulators. Notably, in VPS16, the shared subunit of both the CORVET and HOPS complex, showed the strongest depletion in our screen among the aforementioned regulators (van der Kant et al, 2015). Likewise, we identified large parts of the GARP complex, which, together with the COG complex, forms the Golgi machinery associated with phagocytosis (Pérez-Victoria et al, 2010), and large parts of the retromer complex, an apparatus responsible for the recycling of organelles in the cell (Haft et al, 2000).

*MAGT1/SLC58A1*, a gene whose loss-of-function mutations cause X-linked immunodeficiency with magnesium defects (XMEN; OMIM: 300853) (Watson et al, 2022), significantly scored together with other parts of the OST complex as phagocytosis regulators in our screen and was previously found as host cell regulators of *Legionella pneumophilia* (Jeng et al, 2019). Either MAGT1/SLC58A1 or its paralog TUSC3/SLC58A2 can assist the STT3B–OST complex during post-translational N-glycosylation in the endoplasmic reticulum (Harada et al, 2019; Ravell et al, 2020). However, since *TUSC3* is not expressed in immune cells, THP-1 cells rely only on *MAGT1* (Watson et al, 2022) and indeed, our screen scored only *MAGT1* but not *TUSC3* as a phagocytic regulator. As noted, MAGT1/SLC58A1 is also a transporter of magnesium ions ($Mg^{2+}$), a metal that is recognized recently as being crucial for the regulation of immune response (Maier et al, 2021). Therefore, both functions of SLC58A1, its facilitating role in the STT3B–OST complex, and the magnesium transport function, could play a crucial role in phagocytosis. Since we additionally identified the accessory proteins (DDOST and RPN2) and the catalytic subunits (STT3A and STT3B) of the OST complex as hits, we conclude that N-linked glycosylation plays an important general role in phagocytosis and consequently speculate that dysregulated phagocytosis might be a factor in XMEN disease (Ravell et al, 2020). How exactly N-linked glycosylation affects phagocytosis will require dedicated investigations.

Beyond the hits described so far, numerous uncharacterized genes that had never been previously linked to phagocytosis were scored in our screen. This valuable resource is included in the dataset associated with this study (Table S1). A detailed validation of the identified genes will undoubtedly represent a worthwhile follow-up of the work presented here.

---

PhagoNeg, PhagoEarly, and PhagoLate of a phagocytosis assay performed with cells treated with or without 10 μM GC7. Data are mean ± 95% confidence interval from three technical replicates, representative for two independent experiments. **$P$ ≤ 0.01, ***$P$ ≤ 0.001, and ****$P$ ≤ 0.0001; by Welch's $t$ test. **(E, F)** Schematic representation of the cDNA of *DHPS* overexpressed in cell lines validated in (F). PAM indicates the protospacer-adjacent motif compatible with Cas9 detection, isgDHPS-1 marks the binding site of the respective sgRNA, and N173S, D243A, and ΔY305-I306 indicate mutations introduced in the corresponding DHPS cDNAs. **(F)** Western blot analysis of the expression of endogenous DHPS (*DHPS*[WT]), overexpressed WT or mutated DHPS (*DHPS*-cDNA) and hypusine levels (hypu-eIF5A) in doxycycline-inducible THP-1 knock-out cells grown either with or without doxycycline. The ratio of expression of the respective *DHPS*-cDNA compared to the expression of the reference (*DHPS*-cDNA[−], lane 1) is indicated below the DHPS blot. Asterisk denotes a non-specific band. **(G)** Schematic showing the key steps and enzymes involved in the hypusine axis. K50 highlights the lysine on eIF5A that undergoes the hypusine modification. **(F, H)** Barplot showing the quantification of the PhagoNeg, PhagoEarly, and PhagoLate of phagocytosis assays performed with the cell lines described in (F). Data are mean ± 95% confidence interval from three technical replicates, representative for two independent experiments. ns > 0.05, **$P$ ≤ 0.01, ***$P$ ≤ 0.001, and ****$P$ ≤ 0.0001; by Welch's $t$ test.

To our surprise, however, the top depleted gene in our screen was *DHPS*, along with its two interacting proteins DOHH and eIF5A, which both scored with very high significance. eIF5A is one of the 20 most abundant proteins in proliferating cells (Hukelmann et al, 2016) and therefore, it is not unexpected that it has been implicated in many cellular functions such as translational elongation (Saini et al, 2009), nucleoplasmic transport (Rosorius et al, 1999), cell proliferation, and as a modulator for mRNA decay (Valentini et al, 2002). DHPS generates the post-translational modification deoxyhypusine by conjugation of the aminobutyl moiety of spermidine on the lysine of eIF5A which is then matured to hypusine by DOHH (Park et al, 1981). Over the years, elegant work has led to the structure of the nuclear shuttle for hypusinated eIF5A (Xpo4) (Aksu et al, 2016), the description of RNA-binding properties for hypusinated eIF5A (Xu et al, 2004), and reports on patients with mutations in *eIF5A* (Faundes et al, 2021), *DHPS* (Ganapathi et al, 2019), or *DOHH* (Ziegler et al, 2022) that all showed neurodevelopmental phenotypes. Although until recently (Lindner et al, 2021), the hypusine axis was never identified in the context of phagocytosis, several previous studies suggested its implication in vesicular trafficking, cytoskeleton organization, and immune cell regulation; all processes highly relevant for phagocytosis. Already in 1952, spermine was found as an anti-mycobacterial natural agent (Hirsch & Dubos, 1952) and later, as an inhibitor for cytokine synthesis in human mononuclear cells (Zhang et al, 1997). Yeast studies found a synthetic lethal interaction between *eIF5A* and *YPT1* (the yeast ortholog of mammalian rab genes, coordinators of endocytosis) (Clague, 1998; Frigieri et al, 2008) and the regulators for yeast cell polarity and actin nucleation (*PKC1*, *ZDS1*, *GIC1* [human ortholog *CDC42*], as well as *PCL1* and *BNI1*) as suppressors for eIF5A depletion (Zanelli & Valentini, 2005). Like the bacterial ortholog EFP (Doerfel et al, 2013), eIF5A was found to assist the translation of polyproline stretches in mammalian proteins (Gutierrez et al, 2013; Ude et al, 2013; Barba-Aliaga et al, 2021) that are especially enriched in proteins of the actin/cytoskeleton, RNA splicing/turnover, DNA binding/transcription, cell signaling (Mandal et al, 2014), and in collagen. Furthermore, whereas the DOHH homolog nero and eIF5A showed to be required for autophagy in drosophila (Patel et al, 2009), eIF5A was found in human cells to regulate the translation of TFEB and therefore also regulate autophagy (Zhang et al, 2019). In addition, eIF5A was shown to assist with the co-translational translocation of proteins in the ER (Rossi et al, 2014), and ablation of eIF5A triggers ER stress in mammalian cells (Mandal et al, 2016). Recently, the hypusine axis was described in the context of macrophage respiration (Puleston et al, 2019), and as the effector of the macrophage inflammatory state in adipose tissue (Anderson-Baucum et al, 2021). Finally, mice lacking hypusine are more susceptible to *Heliobacter pylori* and *Citrobacter rodentium* infections (Gobert et al, 2020).

Since we identified the whole hypusine axis and eIF5A as the unique carriers of hypusine, we hypothesized that the hypusinated form of eIF5A must play a significant role in phagocytosis. Our experiments confirmed that the absence of hypusine, either by blocking DHPS, the key enzyme for hypusination, pharmacologically with GC7 or by a knock-down of *DHPS* with Cas9 indeed led to reduced phagocytosis. Remarkably, we could further show that this defect in phagocytosis also occurred when we replaced the endogenous version of DHPS in THP-1 cells with recessive rare variants of mutated DHPS found in patients (N173S and ΔY305-I306) with neurodevelopmental disorders or with a DHPS variant mutated in the predicted binding site of spermidine (D243A).

In summary, our screen functionally outlined the basic phagocytic machinery to involve several hundred gene products organized in at least two dozen protein complexes. Phagocytosis as a long and complex process functionally requires the contribution of several subprocesses, such as cytoskeletal mechanical and membrane-uptake machinery involved in engulfing and internalizing the foreign material in the phagosome, trafficking of the phagosome, fusion with the lysosome followed by acidification and clearance, and recycling to the plasma membrane. Here, publicly available PPI data and complex annotation allowed us to assign genes to cellular functions, at least in those instances where the annotation of cellular protein complexes was unequivocal. Our work offers an improved understanding of how cargo travels through the endo-lysosomal system and we anticipate that this could in turn direct therapeutics better to their proposed site within the cell, such as antibody–drug conjugates (Tsui et al, 2019) or Cas9 (Lino et al, 2018). In addition, it might help identify new targets for interventions in diseases caused by dysfunctional autophagy or endo-lysosomal systems such as the neurodegenerative diseases Alzheimer's disease, Parkinson's disease, or Huntington's disease. Because the screen presented in this work is based on a single reporter, further similar screens with different substrates will help to further refine the network of phagocytosis regulators. Ultimately, we believe that mining and integrating publicly available PPI data with reporter-based genome-wide genetic screens showcased in this study offers a powerful and widely applicable approach for genetically mapping complex biological processes.

# Materials and Methods

### Cell lines

THP-1, U937, and HEK293T cells were purchased from ATCC. THP-1 and U937 were maintained in RPMI1640. HEK293T was maintained in DMEM. All media were supplemented with 10% FBS and antibiotics (100 U/ml penicillin, 100 mg/ml streptomycin), all either from Gibco or Sigma-Aldrich. Cell lines were grown at 37°C in 5% $CO_2$.

### Phagocytosis assay

Phagocytosis assay was conducted as previously described (Colas et al, 2014). Reporter beads were generated by opsonization of dual-colored 1.75-$\mu$m latex beads (Fluoresbrite carboxylated 1.75 $\mu$m microspheres [yellow-green: 441-nm excitation, 486-nm emission, cat no. 17687-5; Polyscience]) in 50% human male AB serum in PBS for 16 h at 4°C under permanent rotation. Then beads were washed twice with PBS and labeled with 2 $\mu$g/ml pHrodo-Red SE in PBS (cat no. P36600; Thermo Fisher Scientific) for 30 min at room temperature under constant agitation. Reporter beads were then washed with PBS and adjusted to a final concentration of $1 \times 10^9$ beads per ml in PBS.

For phagocytosis assays that were performed on individual cell lines, PMA-differentiated cells (THP-1 or U937) were seeded on 12-well cell culture-coated dishes ($1 \times 10^6$ cells per well). Reporter beads were then added to the cells at an MOI of 10 for 3 h. Cells were then transferred onto ice, washed three times with ice-cold PBS, detached by scraping with a soft-rubber cell scraper, and analyzed on an LSR Fortessa II cytometer controlled with FACSDiva (BD), analyzed with FlowJo software (v.10), and visualized with Python project version 3.8.8 (Python Software Foundation. Available at http://www.python.org) with pandas (Reback et al, 2021) (1.2.4), numpy (1.20.1), matplotlib (Hunter, 2007) (3.3.4), seaborn (Waskom, 2021) (0.11.1), and statannotations (0.4.4). Live cells were gated based on FSC-A and SSC-A, followed by FSC-A and FSC-H, and then SSC-A and SSC-H gating to select single cells. The number of pre-gated cells was equal across samples (30,000 pre-gated cells were collected in each sample).

For the Cytochalasin D (cat no. BML-T109-0001; Enzo) and Bafilomycin A1 (cat no. BML-CM110-0100; Enzo) control assays, cell media were supplemented with the compound 30 min before the addition of the reporter beads and added during the whole assay at final concentrations of 8 $\mu$M and 200 nM, respectively.

For the GC7 (cat no. 259545; EMD Millipore)-treated phagocytosis assays, cell media were supplemented during the entire differentiation and seeding period with 10 $\mu$M of the compound.

## Genome-wide CRISPR-Cas9/KO screening and cell sorting

For screening, the genome-wide CRISPR-Cas9/KO Toronto Knockout version 3 library from Hart and team (Hart et al, 2017) (Addgene no. 90294), cloned into the one-vector system (lentiCRISPRv2 carrying Cas9 and sgRNA expression on the same vector), was used. The library consists of 70,948 guides, targeting 18,053 protein-coding genes with four sgRNAs each and 142 control sgRNAs. Viral particles were produced by transient transfection of HEK293T cells with the library and packaging plasmids pMD2.G (Addgene no. 12259) and psPAX2 (Addgene no. 12260) using PEI (Sigma-Aldrich). 8 h post-transfection, the medium was exchanged to RPMI1640 supplemented with 10% FBS. 72 h after transfection, the viral supernatant was collected, filtered (0.45 $\mu$m), and stored at −80°C until further use. THP-1 cells were infected in quadruplicates with this viral supernatant (supplemented with 8 $\mu$g/ml protamine sulfate) at library coverage of 3000× and a MOI of around 0.3, allowing a single-viral integration event per cell. Infected cells were then selected with 2 $\mu$g/ml puromycin for 8 d, followed by 9 d of growth in puromycin-free medium. Afterward, per replicate $750 \times 10^6$ infected and recovered THP-1 cells were differentiated on 15 cm cell culture-coated dishes ($20 \times 10^6$ cells per dish). For differentiation, the cells were kept 48 h in RPMI1640 supplemented with 10% FBS and 10 nM PMA (Sigma-Aldrich), followed by 24 h of growth in RPMI1640 supplemented with 10% FBS without PMA (Fig S1). Phagocytosis assay was performed as described in the section "Phagocytosis assay" using dual-colored opsonized latex beads. ~300 million cells (>3,000× coverage) were used as the input for each phagocytosis assay. All cells were then sorted on a BD FACS Aria II, collecting phagocytosis-positive (PhagoLate) and phagocytosis-negative (PhagoNeg) populations. Genomic DNA was extracted using the QIAamp DNA Mini Kit (QIAGEN) and the sgRNA-containing cassettes were amplified with a one-round PCR approach following the method

of Joung and team (Joung et al, 2017). Amplified samples were then sequenced on a HiSeq2000 (Illumina) at the Biomedical Sequencing Facility (BSF at CeMM, https://biomedical-sequencing.at) and analyzed by a custom analysis pipeline (see below).

## Analysis of CRISPR screens

Sequences of sgRNAs were extracted from NGS reads, matched versus the original sgRNA library index, counted using an in-house Python script, and their abundance assessed using a two-step differential approach. First, differential abundance of individual sgRNAs was calculated using DESeq2 (Love et al, 2014). Afterward, sgRNAs were sorted by adjusted *P*-value and aggregated using the gene set enrichment algorithm (Subramanian et al, 2005; Korotkevich et al, 2016 Preprint; Kuleshov et al, 2016). Essential genes were evaluated by using the same two-step differential approach, comparing sgRNA abundance in undifferentiated cells collected on the day of differentiation to the plasmid stock of the sgRNA library used for virus generation (Fig S1).

## Protein–protein interaction mapping of genetic hits

To probe for physical interactions between the 716 top hits of the genetic screen, a PPI network was generated. The protein interactions were extracted from BioPlex 3.0 (for HEK293T and HCT116, downloaded on 20.04.2022) (Huttlin et al, 2021), whereas only the protein interactions between the genes identified as high scoring within the genetic screen were considered. For the extraction of protein interactions, all bait–prey interactions reported in BioPlex 3.0 were considered, resulting in a PPI network with 490 proteins (69% of all genetic hits) with 726 interactions. The network was further organized into protein complexes and modules by annotating all proteins within the network with complexes reported in the CORUM-core (the comprehensive resource of mammalian protein complexes, CORUM September 2018 release) (Giurgiu et al, 2019). Within CORUM, homo dimers were removed. 246 proteins (50% of all) in the interaction network were reported as a subunit in protein complexes. Network visualization and ordering were conducted in Cytoscape (v.3.8.0). As a first step, all duplicated edges were removed as the directionality obtained by reciprocal AP-MS is not of relevance for the visualization of protein complexes within the network. The interactions were grouped into functional modules by following the association to protein complexes, whereas if a protein is mapped to multiple complexes, which is often the case as the CORUM-core contains complex identifiers with partially overlapping subunits, the complex with the highest completeness was considered. The selection of interactions and mapping of complexes were prepared with the statistical software R (v.4.1.3).

## Plasmids and cloning

Knock-out cell lines were generated using the lentiCRISPRv2 CRISPR-Cas9 knock-out vector (Addgene no. 52961). Inducible knock-out cell lines were produced with the TLCV2 CRISPR-Cas9 backbone (Addgene no. 87360) that includes a doxycycline-inducible Cas9-2A-eGFP cassette. In short, for each gene, sgRNAs were designed using the CHOPCHOP prediction tool (Labun et al, 2019) unless indicated otherwise. Oligos, containing BsmBI-compatible overhangs were annealed and cloned into lentiCRISPRv2 using Golden Gate assembly.

**Cloned oligonucleotides were as follows (5′ to 3′ orientation).**

| sgWASF2-1 | caccgTAACGAGGAACATCGAGCCA | |
|---|---|---|
| sgWASF2-2 | caccgTCGGTCGACCCTCTCAGCAA | |
| sgWASF2-3 | caccgTTGGTGGTATCAGAAAGCGG | |
| sgWASF2-4 | caccgAATGCGACGAGACAAGATGG | |
| sgiDHPS-1 | caccgGTGCGCGGTAATTCACACCG | From TKOv3 library |
| sgiDHPS-4 | caccgTAAGTGCGGGACTAAACACA | From TKOv3 library |
| EXOC1sg1 | caccgCTACCACAGCAAGATCTCGA | From TKOv3 library |
| EXOC1sg2 | caccgGATATGCCAAGCTGATGGAG | From TKOv3 library |
| COG6sg1 | caccgGATGAAATGAGTCTTCTCCG | From TKOv3 library |
| COG6sg2 | caccgTGTGACACTGAAGGCTGCAA | From TKOv3 library |
| MAGT1sg1 | caccgATAACGGAGTAATTTCTCGG | From TKOv3 library |
| MAGT1sg2 | caccgCAGCTGAGCAGATTGCCCGG | From TKOv3 library |
| ARPC2sg1 | caccgGGTGAACAACCGCATCATCG | From TKOv3 library |
| ARPC2sg2 | caccgGAAAGACACAGACGCCGCTG | From TKOv3 library |
| COMMD3sg1 | caccgCTTGAAACATATCGACCCAG | From TKOv3 library |
| COMMD3sg2 | caccgGAGGAGAAGCGTGAAGGCGT | From TKOv3 library |
| COPS3sg1 | caccgCATGATATGACTGACCGCCA | From TKOv3 library |
| COPS3sg2 | caccgGATGGGATAAGTTCTTCGCA | From TKOv3 library |
| LAMTOR5sg1 | caccgGCTCATCTGACAGGGTCCCG | From TKOv3 library |
| LAMTOR5sg2 | caccgGAAACACGATGGCATCACGG | From TKOv3 library |
| STK11sg1 | caccgTGTATAACACATCCACCAGC | From TKOv3 library |
| STK11sg2 | caccgGGCACTGCACCCGTTCGCGG | From TKOv3 library |
| VPS52sg1 | caccgTTGTGACCTGTACTTCAGCA | From TKOv3 library |
| VPS52sg2 | caccgGAAATCGCCAGGCAGTTCGG | From TKOv3 library |
| KXD1sg1 | caccgTGAGCACCCACCTGATACGG | From TKOv3 library |
| KXD1sg2 | caccgGCGGCCGCAGAAGACCCTCG | From TKOv3 library |
| VPS35sg1 | caccgTAGACAAGACATGCCTTCAG | From TKOv3 library |
| VPS35sg2 | caccgATTCTGGTGTAACTCAGCAC | From TKOv3 library |
| P4HA1sg1 | caccgGGATACAGATACCATCTCAA | From TKOv3 library |
| P4HA1sg2 | caccgTGTGGATGGAACAAGCCCTA | From TKOv3 library |
| VHLsg1 | caccgGCGATTGCAGAAGATGACCT | From TKOv3 library |
| VHLsg2 | caccgTGTCCGTCAACATTGAGAGA | From TKOv3 library |
| VPS16sg1 | caccgGAAATATGAGCTGTACAGCA | From TKOv3 library |
| VPS16sg2 | caccgGCAGTGGAAGAGTGGACCCG | From TKOv3 library |
| PSMD13sg1 | caccgGAACCGATGTCACACCAGGA | From TKOv3 library |
| PSMD13sg2 | caccgGCAGGAGAGAGCCTTCACGC | From TKOv3 library |
| KIAA0196sg1 | caccgGTAGAGAATCACGTACAGCA | From TKOv3 library |
| KIAA0196sg2 | caccgACTGCAATGTTGCCATCCGA | From TKOv3 library |
| FAM96Asg1 | caccgAGAGTCGCCAAAGAGCAATG | From TKOv3 library |
| FAM96Asg2 | caccgGATAAATGACAAAGAGCGAG | From TKOv3 library |
| ATP6AP2sg1 | caccgAGGAGAGCGGATCCCAGACG | From TKOv3 library |
| ATP6AP2sg2 | caccgGAGCACGACAAACACAGCCA | From TKOv3 library |
| WDR24sg1 | caccgGTGGCGCCAGGCAATTCCCG | From TKOv3 library |
| WDR24sg2 | caccgGTTCTCAAAGGTGGAGGCGA | From TKOv3 library |

Non-codon–optimized cDNA for *WASF2* was synthesized and cloned into the ENTR–Twist–Kozak vector by Twist. Mutations in the PAM site of *WASF2*-sgRNA1, *WASF2*-sgRNA2, and *WASF2*-sgRNA4 were introduced to make the cDNA resistant against those sgRNAs, using NEB Q5 site-directed mutagenesis kit. After sequence verification, the cDNA was cloned into pRRL-based lentiviral expression plasmids generated in our laboratory (Bigenzahn et al, 2018), which contain a STOP codon and a BlastR-resistance cassette, using the Gateway cloning system (Thermo Fisher Scientific).

DHPS (HsCD00295718, Harvard Medical School plasmid repository), eIF5A (HsCD00041224, Harvard Medical School plasmid repository), and DOHH (HsCD00372432, Harvard Medical School plasmid repository) plasmids were received, and point mutations to make the cDNA resistant against sgRNAs or patient variant mutations were introduced using the NEB Q5 site-directed mutagenesis kit. They were cloned into the same pRRL-based lentiviral expression plasmids as mentioned above, which contained this time, however, either an HA-tag or a STOP codon on the C-terminus and again a BlastR resistance cassette, using the Gateway cloning system (Thermo Fisher Scientific). Rescue experiments were conducted by using the cDNAs resistant to the sgRNAs targeting the endogenous genes.

## Lentiviral transduction

Knock-out cell lines and cDNA-overexpression cell lines were generated by lentiviral transduction. Therefore, HEK293T was transfected with the respective lentiviral vectors and packaging plasmids pMD2.G (Addgene no. 12259), and psPAX2 (Addgene no. 12260) using PEI (Sigma-Aldrich). 16 h post-transfection, the medium was exchanged to RPMI1640 supplemented with 10% FBS. 72 h after transfection, the viral supernatant was collected, filtered through 0.45 $\mu$m polyethersulfone filters (GE Healthcare), supplemented with 8 $\mu$g/ml protamine sulfate, and directly added to the respective cells. 24 h after infection, the medium was exchanged, and 48 h after infection, cells were selected with the respective antibiotics.

## Listeria infection assay

*Listeria monocytogenes* strain 10403S (PMID: 3114382) was grown in Brain–Heart infusion medium overnight at 37°C. The infection assay was performed on individual PMA-differentiated U937 cell clones, knocked out for *WASF2,* and rescued either with a *WASF2*-cDNA (depicted in Fig 3G) or with a mock cDNA as control. A day before infection, $3.0 \times 10^5$ cells were seeded on sterile glass coverslips laid at the bottom of the wells of a 24-well plate in the RPMI1640 medium supplemented with 10% FCS and containing no antibiotics. On the day of the infection, bacteria were washed with PBS and incubated at a concentration of $8.0 \times 10^9$ b/ml with 50 $\mu$M of the green fluorophore CFSE for 20 min at 37°C under shaking. Bacteria were then washed with PBS and used to infect the U937 cells at an MOI of 10. After one hour of infection, the cells were gently washed twice with a warm cell culture medium. For the next two remaining hours, the cells were incubated in the cell culture medium containing 20 $\mu$g/ml of gentamycin to kill the remaining extracellular

bacteria. The infection was terminated by washing the cells with PBS and fixing them with 4% paraformaldehyde (Alfa Aesar) for 15 min. Cells were then washed with PBS and permeabilized with 0.1% of Triton X-100 (Roth) in PBS for 5 min. After another washing step with PBS, cells were blocked with 2% FCS in PBS for 30 min. Actin was then stained by incubating the cells with Phalloidin coupled to the fluorophore Alexa647 (Life Technology). Finally, the cells were washed with PBS and mounted in Prolong Diamond containing DAPI (Molecular Probe) to stain genomic DNA. All staining steps were carried out at room temperature in the dark. Infected cells were imaged with the confocal microscope LSM 700 (ZEISS) using 405, 488, or 639 nm laser lines. Images were processed by the ZEN Software 2009 (ZEISS). A minimum of 200 cells were counted for each technical replicate. Then, a representative image of each technical replicate was chosen, and the total amount of bacteria quantified without further normalization. Cells were seeded at equal density. Thanks to the gentamycin protection assay, all bacteria detected were considered as phagocytosed. Green bacteria were considered still enclosed inside the phagolysosome, whereas yellow bacteria (CFSE and actin colocalized) had successfully escaped the phagolysosome. Quantified bacteria were analyzed and visualized with Python project version 3.8.8 (Python Software Foundation available at http://www.python.org) with pandas (Reback et al, 2021) (1.2.4), numpy (1.20.1), matplotlib (Hunter, 2007) (3.3.4), seaborn (Waskom, 2021), (0.11.1), and statannotations (0.4.4).

## Gene set enrichment analysis

GO enrichment analysis for "Cellular components" GO terms and biological processes GO terms was performed via GSEA (Subramanian et al, 2005) and Enrichr (Kuleshov et al, 2016) using the Python package GSEApy (version 0.10.8, https://github.com/zqfang/GSEApy). Gene sets are described in the corresponding figure legends.

## Cell lysis and immunoblotting

Cell pellets were lyzed in RIPA-lysis buffer (20 mM TRIS–HCl, pH 7.5, 150 mM NaCl, 1 mM EDTA, 1 mM EGTA, 1% NP40, 1% sodium deoxycholate, 10 mM NaF, and 0.1% SDS), supplemented with EDTA-free protease inhibitor (Roche) for 15 min on ice, followed by 20,600$g$ at 4°C for 15 min to separate from cellular debris. Protein concentration was then measured with BCA (Pierce) or Bradford (Bio-Rad), Laemmli buffer was added, and samples were separated by SDS–PAGE and transferred to nitrocellulose membranes (GE Healthcare). Then, the membranes were blocked for unspecific binding in 5% milk in TBS-T followed by incubation with indicated antibodies. After incubating in peroxidase-coupled secondary antibodies, the membranes were developed using the ECL Western blot system (Thermo Fisher Scientific).

Bands in Fig 4F were quantified using Image Lab 6.1 (Bio-Rad). First, the intensity of the *DHPS*-cDNA band and GAPDH band was quantified in each lane. A normalization factor was calculated by dividing the GAPDH band intensity of each lane with the GAPDH band intensity of the first lane (reference). The intensity of the *DHPS*-cDNA band was then multiplied by this normalization factor. Normalized *DHPS*-cDNA expression levels were then compared to normalized *DHPS*-cDNA expression levels in each lane and indicated as ratio (*DHPS*-cDNA/*DHPS*-cDNA$^{(-)}$). All calculated intensities were background-subtracted.

## Antibodies list

The primary antibodies used for Western blot were hypusine (generation see below, 1:20,000), eIF5A (611976, 1:20,000; BD Biosciences), DHPS/DHS (ab190266, 1:1,000; Abcam), DOHH (376929, B-12, 1:500; SantaCruz), HA (3724, C29F4, 1:500), for WASF2 the WAVE-2 (3659, D2C8, 1:1,000; CellSignaling), and GAPDH (47724, 0411, 1:2,000; SantaCruz).

The secondary antibodies used were anti-rabbit HRP (111-035-144, dilution 1:10,000; Jackson ImmunoResearch), anti-mouse HRP (115-035-003, dilution 1:10,000; Jackson ImmunoResearch).

A custom rabbit anti-hypusine antibody was designed based on a 2016-published sequence of context-independent anti-hypusine antibodies (Zhai et al, 2016) (Fig S6A), synthesized, expressed, and purified with Genscript (Fig S6B). In short, the synthesized sequence was cloned into pcDNA3.4 and plasmid DNA prepared. Expi293F cells, grown in a serum-free Expi293F Expression Medium (Thermo Fisher Scientific), at 37°C, 8% $CO_2$ on an orbital shaker, were then transfected according to the manual. On day six, post-transfection cell culture supernatant was collected and used for purification. Therefore, the supernatant was centrifuged, filtered, and then loaded and purified on Monofinity A Resin Prepacked Columns. Eluted antibody was then puffer exchanged, sterilized via a 0.22-$\mu$m filter, and stored in PBS pH 7.2 at –80°C. The concentration was determined with A280 as 4.74 mg/ml. Specificity was validated by Western blot in HEK293T cells overexpressing DHPS/DOHH and either an eIF5A-WT sequence or an eIF5A-K50 variant (Fig 4B), and in THP-1 cells treated with 10 $\mu$M of the DHPS inhibitor GC7 (Fig 4C).

## Gene expression analysis

Libraries compatible with Illumina sequencing were prepared using the QuantSeq 3′ mRNA-seq Library prep kit (Lexogen) according to the manufacturer's instructions. Samples were multiplexed and then sequenced on HiSeq 4000 (Illumina) at the BSF at CeMM. Raw-sequencing reads were de-multiplexed and barcode, adaptor, and quality trimming were performed with cutadapt (https://cutadapt.readthedocs.io/en/stable/). Quality control was then performed using FastQC (http://www.bioinformatics.babraham.ac.uk/projects/fastqc/). The remaining reads were mapped to the GRCh38/h38 human genome assembly using genomic short-read RNA-seq aligner STAR v.2.5. More than 98% of the mapped reads in each sample could be obtained with 70–80% of reads mapping to a unique genomic location. End Sequence Analysis Toolkit was used to quantify the transcripts. We carried out a differential expression analysis using independent triplicates with DESeq2 (v.1.24.0) on the basis of read counts. Exploratory data analysis and visualizations were performed in R-project v.3.4.2 (Foundation for Statistical Computing, https://www.R-project.org/) with Rstudio IDE v.1.2.1578, ggplot2 (v.3.3.0), dplyr (v.0.8.5), readr (v.1.3.1), and gplots (v.3.0.1).

## DepMap comparison

Growth effects induced by knock-out of individual genes recorded for THP-1 cells in our screen were compared with the growth effects recorded in the DepMap CRISPR dataset. Preprocessed CRISPR dependency scores (DepMap 22Q2 Public+Score, Chronos) (DepMap, 2022) were mined for all recorded genes in the cell line THP-1 from DepMap and extracted for essential genes (genes depleted on day 21, $P \leq 0.05$, Table S2) and the 716 CRISPRko hits depleted in PhagoLate versus PhagoNeg. For 1,040 out of 1,041 essential genes and 671 out of 716 CRISPRko hits a matching Chronos score could be identified. For comparing essential genes, Chronos scores of essential genes were compared to all CRISPR-derived dependency scores recorded for THP1 in the DepMap project in a density plot. Likewise, the distribution of growth effects caused across the 716 CRISPRko hits were compared to all other genes recorded in the DepMap dataset (17,384 genes) in a density plot. Data were visualized with Python project version 3.8.8 (Python Software Foundation available at http://www.python.org) with pandas (Reback et al, 2021) (1.2.4), numpy (1.20.1), matplotlib (Hunter, 2007) (3.3.4), and seaborn (Waskom, 2021) (0.11.1).

## "Core essentialome" comparison

Our list of essential genes (genes depleted on day 21, $P \leq 0.05$, Table S2) was compared to the "core essentialome," a list of 1,736 shared essential genes that have been derived from HAP1 and KBM7 cells (Blomen et al, 2015) in a Venn diagram. Data were visualized with Python project version 3.8.8 (Python Software Foundation available at http://www.python.org) with pandas (Reback et al, 2021) (1.2.4), numpy (1.20.1), matplotlib (Hunter, 2007) (3.3.4), and matplotlib_venn (0.11.6).

# Data Availability

Genomics datasets are provided in Tables S1 and S2. The transcriptomics dataset is attached in Table S3. PPI data mined from BioPlex 3.0 and CORUM-core specifically for this study are provided in Tables S4 and S5. DepMap cell line CRISPR scores were obtained from DepMap 22Q2 (https://depmap.org/portal/download/all/).

# Supplementary Information

# Acknowledgements

We thank all members of the Superti-Furga laboratory for feedback, discussions, and reagents; the Max Perutz Labs FACS Facility for sorting; and the Biomedical Sequencing Facility (CeMM/Medical University of Vienna) for next-generation sequencing. In particular, we thank Andrea Garofoli for advice on statistics. We also thank thesis advisor Thijn Brummelkamp for helpful suggestions throughout the project. We thank Kai-Chun Li, Ann-Katrin Hopp, and Andreas Angermayr for scientific discussions and critical reading. CeMM and the Superti-Furga laboratory are supported by the Austrian Academy of Sciences. We further acknowledge the receipt of third-party funds from the European Research Council (ERC) Advanced Grant 695214 GameofGates (P Essletzbichler, V Sedlyarov, G Superti-Furga).

## Author Contributions

P Essletzbichler: conceptualization, data curation, software, formal analysis, supervision, validation, investigation, visualization, methodology, writing—original draft, review, and editing.
V Sedlyarov: data curation, software, formal analysis, investigation, visualization, and methodology.
F Frommelt: data curation, software, formal analysis, investigation, visualization, and methodology.
D Soulat: formal analysis and investigation.
LX Heinz: investigation.
A Stefanovic: investigation.
B Neumayer: investigation.
G Superti-Furga: conceptualization, supervision, funding acquisition, project administration, writing—original draft, review, and editing.

## Conflict of Interest Statement

G Superti-Furga and P Essletzbichler have filed patents based on this work that may lead to commercialization efforts in the future.

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
