## [Reviewer comments · Life Science Alliance]

Life Science Alliance

A genome-wide CRISPR functional survey of the human phagocytosis molecular machinery

Patrick Essletzichler, Vitaly Sedlyarov, Fabian Frommelt, Didier Soulat, Leonhard Heinz, Adrijana Stefanovic, Benedikt Neumayer, and Giulio Superti-Furga

DOI: <https://doi.org/10.26508/lsa.202201715>

Corresponding author(s): Giulio Superti-Furga, CeMM Research Center for Molecular Medicine

Review Timeline:

Submission Date:	2022-09-07
Editorial Decision:	2022-10-20
Revision Received:	2022-12-09
Editorial Decision:	2023-01-16
Revision Received:	2023-01-18
Accepted:	2023-01-19

Scientific Editor: Novella Guidi

Transaction Report:

October 20, 2022

Re: Life Science Alliance manuscript #LSA-2022-01715

Prof. Giulio Superti-Furga
CeMM Research Center for Molecular Medicine
CeMM Research Center for Molecular Medicine of the Austrian Academy of Sciences
Lazarettgasse 14
Vienna 1090
Austria

Dear Dr. Superti-Furga,

Thank you for submitting your manuscript entitled "A genome-wide CRISPR functional survey of the human phagocytosis molecular machinery" to Life Science Alliance. The manuscript was assessed by expert reviewers, whose comments are appended to this letter. We invite you to submit a revised manuscript addressing the Reviewer comments.

Thank you for this interesting contribution to Life Science Alliance. We are looking forward to receiving your revised manuscript.

Sincerely,

B. MANUSCRIPT ORGANIZATION AND FORMATTING:

Reviewer #1 (Comments to the Authors (Required)):

This research study aims to expand the current view of the factors involved in phagocytosis. To do this, Essletzichler et al. introduce an elegant approach for the unbiased identification of the cellular machinery involved in the uptake of large particles, or phagocytosis. The authors perform a sorting screen for THP-1 macrophages that have internalized opsonized 1.75 micron latex beads. The beads serve as a surrogate of common phagocytosis targets such as microbes, cells undergoing apoptosis, and malignant cells. They derive a reporter system and divide macrophages into 3 sub-populations corresponding to non-phagocytic macrophages (no beads), phagoEarly (macrophages that have internalized beads whose pHrodo sensor is not fully activated by the acidic microenvironment following phagolysosome fusion) and phagoLate (indicative of strong pHrodo signal triggered by the acidic environment of the mature phagolysosome). In other words, cells deficient in phagocytosis would be negative for the YG fluorophore, whereas YG-positive cells that arrest at the acidification step would be negative for the pHrodo-Red fluorophore. Having validated that their reporter system accurately captured drug-induced deficits in engulfment and acidification, the authors performed a FACS-based pooled genome-wide CRISPR screen to identify genes that prevented bead uptake and/or acidification. The authors found several bona fide regulators of bead uptake as well as genes that have never been mechanistically associated with phagocytosis.

The authors also present an approach for analysing and interpreting the CRISPR sorting screen result by integrating the hits with proteomics data. Specifically, each hit is associated with known protein complexes and interactions between protein complexes (from the BioPlex interactome and Corum) are taken into consideration when selecting specific hits for downstream validation. This allows the authors to confirm known regulators of phagocytosis (Arp2/3, Wave-2), phagosome trafficking and maturation (components of the vacuolar ATP-Rag, BORC and the interacting exocyst, GARP and retromer complexes), oligosaccharyltransferase complex constituents and the eIF5a hypusination pathway.

The study highlights the power of integrating functional genetic screening data with publicly available protein-protein interaction data.

Major Comments:

- YG positive signal indicates that cells are competent to engulf beads, whereas pHrodo-Red positive signal suggests intact acidification. In Figure 1b, YG signal seems to be discrete, multimodal whereas pHrodo-Red signal seems to be continuous with two major peaks likely representing acidified/non-acidified states. Why is YG signal not continuous? Is this discreteness correlated with bead load, going from 0 (PhagoNeg), to 1, 2, 3, etc (PhagoEarly)? If so, YG signal should become smoother over time and stabilize at around 10 beads per cell, as beads were added at a 10:1 bead-to-cell ratio. The authors should clarify why treatment time and bead-to-cell ratio were optimized to yield ~33% of cells in the PhagoNeg fraction. Under the chosen conditions, roughly ~33% of the cells miss a chance to engulf any beads due to technical constraints, rather than underlying genetic defects induced by CRISPR/Cas9-mediated gene editing, thus making it difficult to discern whether the hits represent true biology or differences in guide abundances governed by stochastic processes, or does the bead uptake by wild type PMA-differentiated THP-1 cells saturate at ~67% at a cost-permitting bead-to-cell ratio under a reasonable time frame?
- The usage of 1.7µm microspheres is more suited to phagocytosis of large particles and small micro-organisms. It is of note that a comparable screen has already been performed (doi: 10.1038/s41588-018-0254-1) in U937 cells, where beads of multiple sizes were employed for phagocytosis. The authors should compare and discuss the intersection of results with this study.
- The authors pursued validations by creating gene knockouts for 18 out of 716 hits, and found that 15 of them displayed the predicted phenotype in terms of bead uptake. The authors should clarify how these genes were selected, especially since recall efficiency (15/18) is quoted to support the validity of the screen.
- Moreover, the 716 screen hits were probed for physical interactions. The screen hits were first filtered to remove essential genes. The authors should clarify how this step was performed as some genes included in the network and pursued in validations are bona fide common essential genes identified by others (<https://depmap.org/portal/>).
- In relation to the comment above. "Despite affecting phagocytosis in our screen, Cdc42 has very high general essentiality and therefore was scored correctly as an essential gene rather than a hit in our screen (Suppl Table 2). This case demonstrated the well-known drawback of knock-out screens for specific processes and subsequent readouts, which typically fail to identify highly essential genes". The authors should clarify how they define gene essentiality, how it was predicted, and whether it is a

qualitative feature of a gene (i.e. essential vs non essential) or falls under a spectrum (i.e. can be very high). If understood correctly, this statement comes in conflict with the filtering step prior to generating the protein-protein interaction network, as essential genes should not have been identified in the first place. Moreover, THP-1 cells have gone through very few doublings before the sorting for phagocytosis was conducted. This implies that the time after infection may not have been sufficient for essential genes to drop-out. In this context, it is unclear how essential genes were filtered out before downstream analysis and confounds some of the statements in the manuscript.

- The authors used latex beads for their research. It is possible that some of the hits are driven by the physicochemical properties of latex beads, which could be different from those of microbes, apoptotic cells, and malignant cells in terms of size, rigidity, and curvature; factors that are known to influence phagocytosis? Further validation would improve the generalizability and impact of the study.
- Validation of hits were only done in THP-1 cells. Therefore, the findings cannot be generalized to "the human phagocytosis molecular machinery," as indicated in the title of the manuscript. We encourage the authors to validate the screen hits in other human monocytic cell lines (both monocytes and macrophages) as well as in primary human monocytes and monocyte-derived macrophages to confirm that the hits are regulating phagocytosis broadly across different models.
- The authors confirm the eIF5a hypusination pathway (doi: 10.1038/s41598-021-92332-7) as a positive regulator of phagocytosis. However, it is not clear mechanistically how it may be affecting phagocytosis. That is, by promoting phagosome-lysosome fusion (as suggested by figure 4h since there is no change in phagoEarly fraction) or by inducing target engulfment and internalization? In addition, it would be helpful to assess the mRNA targets of hypusinated versus unmodified eIF5a and whether they may be implicated in phagocytosis, including phagosome maturation, in order to improve the findings of the manuscript.

Other comments:

- Fig 1b. The Red and YG signals increase with intensity implying potential compensation issues. Have the stains been properly compensated? In addition, have there been any experiments looking at the stability as a function of time for pHrodo Red signal. For how long can pHrodo emit signal in the lysosome before it is degraded? The increased pHrodo signal seems to be increasing with number of internalized beads. Does this imply that certain sub-populations of cells are hyperactive in not only phagocytosing, but also processing the cargo? This could be assessed by comparing YG-beads (no pHrodo) and beads-pHrodo (no YG signal).
- Fig 2. A bit difficult to follow the color scheme of the figure. It would be better to have a 3-color scheme for the log₂(FC) (with 0 being white) with colors designating complex different from the log₂(FC) colors.
- Fig 3a. It would be more informative to show effect size for significant genes rather than FDR or p-value. Also, correct the p-value scale in Figure 3a, as p = 0.04 appears twice, and p {less than or equal to} 0.05 already includes 0.04, 0.02, 0.01 and 0.
- Fig. 3b. We would expect to see the strongest change in the phagoEarly fraction since Arp2/3 is critical for the phagocytic cup formation. The authors should comment on the fact that the observed results in their model do not recapitulate the expected effect of Arp2/3 ablation.
- Fig. 4f. There are several bands associated with DHPS. Do they correspond to different isoforms, PTMs or non-specific binding? As per the KO (lane 1 vs lane 6) it would appear that the top band corresponds to wt DHPS since it is larger in the Dox- compared to Dox+ condition. Quantification will be helpful.
- It is not clear how THP-1 cells were differentiated exactly (how many days of 10nM PMA). There are many protocols for PMA-dependent differentiation of THP-1 cells into macrophages. Some protocols, particularly the ones employing only 10nM of PMA, are associated with a significant number of non-differentiated suspension cells, which will result de facto in a decrease in the fold gene coverage of each gRNA cassette for the CRISPR screen.
- The authors focus on depleted gRNA cassette reads from the phagoLate fraction only to determine positive regulators of phagocytosis. What about the enriched gRNA cassettes in phagoLate? What is the overlap between depleted in phagoLate and enriched in phagoNeg?
- "Interestingly, the knock-out of VPS35, a subunit of the retromer complex, significantly reduced the PhagoNeg and PhagoEarly fraction, but not the PhagoLate fraction, confirming in a functional assessment of the different phagocytosis steps, that VPS35 regulates the uptake of materials to the phagosome and less the later part of phagocytosis." (pp10) The effect size for this hit is not indicated on figure 3a and the effects on phagocytosis for this hit have not been experimentally validated.
- "Through the work presented here, it is possible to ascribe a biological role in phagocytosis to the CCC/Commander complex, likely to be related to the vesicular sorting function of the retromer complex (Figure 2, Figure 3a)." (pp10) Very little/no validation shown for this statement.

Reviewer #2 (Comments to the Authors (Required)):

In this manuscript, entitled "A genome-wide CRISPR functional survey of the human phagocytosis molecular machinery", Essletzichler et al. use a FACS reporter-based pooled genome-wide CRISPR knockout screen to dissect the positive genetic interactors of phagocytosis in a human myeloid cell line. The authors score 716 targets, mapping them elegantly to a protein-protein interaction network, which facilitates the contextualization of single hits in different functional protein clusters and serves

as a landmark to guide a conservative and rational selection of hits for validation. The validation of 18 different targets belonging to the most represented/complete functional complexes with single KO experiments and, in few instances, with add back experiments, confirms overall a successful screening strategy. The large set of targets identified in the genetic screen belong mostly to well-known protein complexes regulating several aspects of phagocytosis, starting from engulfment, and spanning through trafficking and signaling of endosomes and lysosomes, as well as N-linked protein glycosylation. In some cases, in addition to previous findings, this work expanded the number of members of a specific pathway/protein complex identified as in the case of the magnesium transporter MAGT1, and specifically confirmed the emerging role of the eIF5A/hypusination pathway in the context of phagocytosis.

Overall, this work corroborates and expands recent studies on this subject and provides a good example for the visualization and rationalization of functional connections among hits of a genetic screen. The manuscript is written in a clear form and well-presented figures; the phagocytosis assay setup, the screen analysis and the validation experiments are robust enough to support a correlation between screen significance score and the functional role of the hit in phagocytosis.

Even though the threshold set to identify hits has a relative loose stringency, I can still appreciate the biological significance of the candidates to phagocytosis thanks to the very meaningful integration of the average sgRNA scores per gene to the protein interaction map.

Specific comments:

-Can the authors describe how they defined gene essentiality in their screen? For clarity a step-by-step process of analysis should be described and included in methods, including which datasets were compared and the rationale for this choice. It remains unclear how the authors define essentiality and why genes like Rab7A, appears both as hit in their screen (table S1) and in their essentiality list (table S2). Please clarify.

-On the same line of thought, how do the authors explain the discrepancy between their gene essentiality analysis and essentiality scores reported for the same cell line (THP-1) in the database DeepMap (<https://depmap.org/portal/>) for example? For instance, in the case of eIF5A and DHPS the hits do not appear in the essentiality list while they score as essential in DeepMap.

-In Fig 1.b: (I) Can the authors explain or eventually show what was the pre-gating strategy? Were cell singlets pre-gated or was performed any sort of nuclear or viability staining? It is not reported in the methods section and I would recommend for it to be added.

(II) Since the % of cells for each bin (phago-negative/early/late) are used to calculate statistics and infer significance, is the number of the pre-gated cells equal and normalized across samples? (e.g. using a down sampling algorithm).

(II) If only singlets were pre-gated, is the "striped" pattern in the "YG" channel actually showing an increasing, discrete number of beads binding to single cells?

-In Fig 1.c and any other figure reporting this type of analysis, I would advise strongly to plot data relative to each replicate (e.g. as single dots, superimposed) in addition to the error bar/confidence interval. For better interpretation of the data, the authors should specify whether the replicates included are independent biological experiments or technical replicates within the same assay.

-In Fig 2 I believe neither the legend nor text specify what the very big light orange circles represent, unless I am missing it, this information should be included.

-In Fig 3a: While it is visually nice I have a few issues with this heatmap:

(I) each cell represents 3 technical replicates and a total of two biological replicates were performed. Correct? Please clarify this point in the legend and for Fig 3b.

(II) The heatmap shows a statistical difference but not the direction of the difference - so we must assume always that there is only one direction: for the KOs increased % of cells in the "phagoNegative" and "phagoEarly" and decreased in the "phagoLate" compared to control. Is this confirmed for all gene specific KO pools since is not visible in the heatmap?

(III) The incongruence within replicates suggests that KO induced defect in phagocytosis is often of modest entity and therefore in general this assay would have benefited of a larger number of biological replicates which in turn would yield more robust and/or ponderate interpretation of the penetrance of the KO phenotypes. The authors should list in addition to the COPS, P4HA1 and PSMD13 also COG6, ATP6AP2, COMMD3 and any other target that did not reproduce at least in duplicate. If this is correct, the authors should rectify the sentence "In summary, of the genes that we assessed individually in phagocytosis assays, tracking three different stages, 83%..." to reflect either the correct percentage of genes that reproduced in duplicate experiments or a quantitative adjective.

-In Fig 3d: For clarity, the authors should specify in the methods section more details about the image quantification process of the bacterial infection, including number of fields used per experiment, method used for automated counting and whether any sort of normalization was performed (e.g. number of bacteria per cell).

-I recommend that the authors describe the content of the supplementary files, including a legend describing each column

header.

-I cannot help but wonder whether the authors performed the analysis to identify enriched sgRNA in the "phagoLate" fraction as compared to "phagoNegative" or "input" fractions as would possibly provide further insights about negative regulators of phagocytosis as reported in several previous studies, also referenced in this manuscript. Unless this is a reserved subject for a future study, I would suggest including it in this manuscript if available.

Minor:

- please describe in more detail the treatment of cells with PMA, number of days of treatment is lacking.
- Across the manuscript "Welch" instead of Welsh.
- "accumulation of toxic disease proteins" =toxic accumulation of proteins
- "expluded" = excluded
- "amendable" = amenable
- "its novel link to phagocytosis" = while some members of the OST complex are newly associated to phagocytosis, the OST complex has been previously associated with phagocytosis (e.g. in Haney et al.), hence the word novel should be avoided in this context, or the sentence rephrased.
- "abolishe" =abolish

-

Reviewer #3 (Comments to the Authors (Required)):

The manuscript by Essletzichler et al. reports:

- I.the results of a CRISPR-CAS9 screen on the phagocytic activity of THP-1 cells designed to distinguish genes that are required for the internalization step (whose depletion leads to a phago-negative phenotype) from those required for phagosome maturation/fusion with lysosomes (whose depletion leads to a phago-early phenotype);
- II.an integrated analysis of the genetic hits that emerged from the screen with the existing protein-protein interaction data involving those hits, which led to the definition of a comprehensive network of 490 genes controlling the phagocytosis process;
- III.the validation of the top hit from the screen, the hypusine axis.

Overall, the data are of very high quality and fully support the conclusions. However, they also raise some concerns that should be addressed.

- 1.Several of the hits described in the manuscript were identified in previous genome-wide screens on the phagocytosing ability of macrophages. Hence, a part of the data confirms published data, so the authors should reduce the description of confirmatory data to expand and highlight what their study adds to the pre-existing published data.
- 2.One innovative aspect of the screen concerns the discrimination of the phago-negative from the phago-early phenotype, and this aspect should be exploited more. In fact, the authors do not exploit the power of their assay. They could extend the analysis of the hits inducing the phago-early phenotype. Considering that the distinction between early and late phagosome phenotypes is based on the pH of bead-containing organelles, they should assess whether the early-phenotype is due to phagosomes that cannot fuse with properly acidified lysosomes or to phagolysosomes with a less acidic pH.
- 3.In the workflow followed in the manuscript, the PMA treatment (for a time that is not specified) that induces the macrophage differentiation of THP-1 cells has been performed 4 days after the infection of THP-1 cells with lentiCRISPRv2 carrying Cas9 and sgRNA. Thus, one important control that should be performed is to check whether some of the genes that were hits in the phagocytosis assay were indeed affecting the differentiation process of THP-1 into macrophages. This control could be performed by analyzing the markers commonly monitored during this differentiation process (CD86, CD11, CD14).

General comments:

We thank the reviewers for the constructive points of criticism and propose to first address a general point that was requested by all three referees, before addressing the points raised individually.

General Point: Essentiality scores for genes

We fully agree that the previous version of our manuscript did not describe in enough detail how essentiality was assessed in our screen. We now amended and complemented the statements. We performed additional analysis and comparisons to DepMap to explain our approach better. In summary we:

- added a schematic summarizing the workflow of our phagocytosis screen (**Suppl Figure 1**)
- compared our list of “essential genes”, as well as the complete hitlist of our screen to gene effect scores recorded in DepMap (**Suppl Figure 5a,c**) and to the “core essentialome” of Blomen et. al 2015 (**Suppl Figure 5b**)
- improved the method part describing the timeline of the CRISPR screen

As illustrated in **Suppl Figure 1**, we collected cells on the day of differentiation (day 21 post-infection) and isolated gDNA. By comparing sgRNA abundance on day 21 to the sgRNA abundance in the plasmid stock of the library we derived a list of genes that lead to reduced growth. We called genes that were significantly depleted on day 21 “essential genes”. As suggested by the reviewers we then took this list of genes and compared them with growth effects reported in the DepMap CRISPR dataset for THP-1 (DepMap 2022). Therefore, we extracted the DepMap reported growth effect values for our list of 1041 essential genes and compared them in a density plot with the growth effect values of all other genes in the DepMap dataset. As one can now see in **Suppl Figure 5a**, our approach to extracting essential genes compares very well with what was reported in DepMap. Details about the data mining and analysis were added to the method section on **page 27**. Additionally, we intersected our list of essential genes with the “core essentialome”, a list of 1736 shared essential genes we have previously derived from HAP1 and KBM7 cell lines (Blomen et al. 2015) (**Suppl Fig 5b**).

In the manuscript, we used this list of essential genes only to categorize them in the volcano plots as in **Figure 1d**. For the rest of the data analysis and data integration presented in the manuscript of the paper, we proceeded with the complete hitlist of 716 genes (depleted in PhagoLate vs PhagoNeg), contrary to what was written in the previous version of the manuscript. We corrected this in the revised version and apologize for the confusion. We now further explain on **pages 6-7** and **Suppl Figure 5a-c** why we proceeded with the full list of 716 genes. Since we are using a reporter-based FACS sorting approach to screen for enriched and depleted sgRNAs and not a proliferation-based selection approach, we decided that it is reasonable to keep also essential genes within the dataset of phagocytosis regulators. Thank you for providing guidance on this issue.

Point-by-point address to Referee #1

(italics are reviewer comments)

This research study aims to expand the current view of the factors involved in phagocytosis. To do this, Essletzbichler et al. introduce an elegant approach for the unbiased identification of the cellular machinery involved in the uptake of large particles, or phagocytosis. The authors perform a sorting screen for THP-1 macrophages that have internalized opsonized 1.75 micron latex beads. The beads serve as a surrogate of common phagocytosis targets such as microbes, cells undergoing apoptosis, and malignant cells. They derive a reporter system and divide macrophages into 3 sub-populations corresponding to non-phagocytic macrophages (no beads), phagoEarly (macrophages that have internalized beads whose pHrodo sensor is not fully activated by the acidic microenvironment following phagolysosome fusion) and phagoLate (indicative of strong pHrodo signal triggered by the acidic environment of the mature phagolysosome). In other words, cells deficient in phagocytosis would be negative for the YG fluorophore, whereas YG-positive cells that arrest at the acidification step would be negative for the pHrodo-Red fluorophore. Having validated that their reporter system accurately captured drug-induced deficits in engulfment and acidification, the authors performed a FACS-based pooled genome-wide CRISPR screen to identify genes that prevented bead uptake and/or acidification. The authors found several bona fide regulators of bead uptake as well as genes that have never been mechanistically associated with phagocytosis.

The authors also present an approach for analysing and interpreting the CRISPR sorting screen result by integrating the hits with proteomics data. Specifically, each hit is associated with known protein complexes and interactions between protein complexes (from the BioPlex interactome and Corum) are taken into consideration when selecting specific hits for downstream validation. This allows the authors to confirm known regulators of phagocytosis (Arp2/3, Wave-2), phagosome trafficking and maturation (components of the vacuolar ATP-Rag, BORC and the interacting exocyst, GARP and retromer complexes), oligosaccharyltransferase complex constituents and the eIF5a hypusination pathway.

The study highlights the power of integrating functional genetic screening data with publicly available protein-protein interaction data.

We thank the reviewer for the detailed assessment of our manuscript and positive judgment. We are grateful for the insightful comments, which we fully addressed point-by-point here below and in the revised manuscript.

Major Comments:

1. YG positive signal indicates that cells are competent to engulf beads, whereas pHrodo-Red positive signal suggests intact acidification. In Figure 1b, YG signal seems to be discrete, multimodal whereas pHrodo-Red signal seems to be continuous with two major peaks likely representing acidified/non-acidified states. Why is YG signal not continuous? Is this discreteness correlated with bead load, going from 0 (PhagoNeg), to 1, 2, 3, etc (PhagoEarly)? If so, YG signal

should become smoother over time and stabilize at around 10 beads per cell, as beads were added at a 10:1 bead-to-cell ratio.

We thank the referee for the careful observation. Indeed, the YG signal represents the bead load within the individual cells in a discrete manner. We now described this feature of the assay in the figure legend of **Figure 1b**.

While it is true, that we added beads to the cells at a bead-to-cell ratio of 10:1, we do not see that the YG signal stabilizes at around 10 beads per cell. Instead, phagocytosis stabilizes at the endpoint of our assay ~3 hours and cells seem to have a pre-defined maximum number of beads that can be phagocytosed. We addressed the phagocytosis assay in more detail in the next point.

1. The authors should clarify why treatment time and bead-to-cell ratio were optimized to yield ~33% of cells in the PhagoNeg fraction. Under the chosen conditions, roughly ~33% of the cells miss a chance to engulf any beads due to technical constraints, rather than underlying genetic defects induced by CRISPR/Cas9-mediated gene editing, thus making it difficult to discern whether the hits represent true biology or differences in guide abundances governed by stochastic processes, or does the bead uptake by wild type PMA-differentiated THP-1 cells saturate at ~67% at a cost-permitting bead-to-cell ratio under a reasonable time frame?

We thank the referee for raising this very important question. Indeed, the incubation period of our phagocytosis assay was optimized for two parameters: First, the endpoint was chosen such that further incubation did not lead to significant further phagocytosis. We now addressed this point in the manuscript on **page 6**. Second, the bead-to-cell ratio was optimized to yield around 1/3 of cells in PhagoNeg and 2/3 of cells in the phagocytosis- and acidification-positive fractions (PhagoEarly and PhagoLate). The main reason for this optimization is, that in order to perform PCR followed by NGS and analysis for depleted sgRNAs, a sufficient amount of genomic DNA has to be extractable from the sorted cells. To illustrate why we chose 3 hours as an endpoint for the phagocytosis assay, we performed a time course phagocytosis assay with WT PMA-differentiated THP1 and our reporter beads and added this new data to the revised manuscript (**Suppl Figure 2a**). It shows that while there is significantly more phagocytosis from the 2 hours to the 3 hours time point, there is not significantly more phagocytosis from the 3 hours to the 4 hours time point.

2. The usage of 1.7um microspheres is more suited to phagocytosis of large particles and small micro-organisms. It is of note that a comparable screen has already been performed (doi: 10.1038/s41588-018-0254-1) in U937 cells, where beads of multiple sizes were employed for phagocytosis. The authors should compare and discuss the intersection of results with this study.

We thank you for your suggestion, are fully aware of the great work of Haney and colleagues (Haney et al. 2018) and therefore mention and compare our work several times in the manuscript to their datasets. Although the two projects are different in terms of cell lines (THP-1 vs U937 and mouse cells), screening approach (FACS vs magnetic beads),

reporter/substrate, genetic library, and gene annotation, which makes the comparison more challenging, we believe that we found an elegant and informative way to address the reviewers question and compare the two datasets. For this, we looked for overlaps of our 716 phagocytic regulators depleted in PhagoLate vs PhagoNeg compared to all significant depleted genes (FDR < 5%) extracted from the study of Haney et al.:

In an upset plot comparing these two datasets, we can see that 79 out of 271 (29%) genes identified by Haney et al., are also part of our dataset. On the left, the plot indicates the size of the two datasets used for the comparison (the 716 hits of this study, 271 hits from Haney et al.).

In the second upset plot, we now analyzed intersections of our dataset with all the specific hits for the different substrates derived from the screens of Haney and colleagues. First, we can see that 15 genes are common to all datasets derived from beads as a reporter, indicating that our dataset contains all except one gene that is common to regulate the uptake of beads of 4 different types. Furthermore, our dataset contains the 6 genes that Haney et al. identify in all gene lists and in addition also the 6 genes specific to the

“Zymosan” screens. Lastly, our dataset has the largest intersection (17 genes) with the dataset derived with RAW264.7 mouse cells. In short, while at first sight the 29% of all regulators identified by Haney et al. can be found in our dataset does not look very impressive, the analysis shows that the datasets of Haney et al. are also very different among each other and are in general smaller in size (54-140 genes). At the same time, our dataset indeed has by far the largest intersection with the different gene lists. This shows that both studies are indeed complementary. While Haney et al. cleverly set up a cost-effective approach (magnetic separation) to screen in their primary screens many substrates and fish out highly specific regulators for different substrates, we on the other side used the more sensitive FACS sorting approach already for our large primary screen. While FACS sorting is more time and cost intense, it allowed us to derive an extensive list of phagocytic regulators that we then ordered by strict integration of these hits using publicly available protein-protein interaction data.

3. The authors pursued validations by creating gene knockouts for 18 out of 716 hits, and found that 15 of them displayed the predicted phenotype in terms of bead uptake. The authors should clarify how these genes were selected, especially since recall efficiency (15/18) is quoted to support the validity of the screen.

We thank the referee for raising an important question. Our rationale for choosing the targets for arrayed validation was to make use of the network of protein complexes. Therefore, we chose complexes for which we identified a high coverage of subunits or complex completeness directly from our genetic hits. We now extended the explanation in the text (**page 8**) and highlighted the validation path of the chosen complexes in Figure 2 in light orange. There are several additional aspects that can also be considered validation, such as the hit frequency of the same complexes and coverage of established known pathways and machines.

4. Moreover, the 716 screen hits were probed for physical interactions. The screen hits were first filtered to remove essential genes. The authors should clarify how this step was performed as some genes included in the network and pursued in validations are bona fide common essential genes identified by others (<https://depmap.org/portal/>).

We thank the referee for this very important assessment. We addressed this problem in detail as a larger general point above.

5. In relation to the comment above. "Despite affecting phagocytosis in our screen, Cdc42 has very high general essentiality and therefore was scored correctly as an essential gene rather than a hit in our screen (Suppl Table 2). This case demonstrated the well-known drawback of knock-out screens for specific processes and subsequent readouts, which typically fail to identify highly essential genes". The authors should clarify how they define gene essentiality, how it was predicted, and whether it is a qualitative feature of a gene (i.e. essential vs non essential) or falls under a spectrum (i.e. can be very high). If understood correctly, this statement comes in conflict

with the filtering step prior to generating the protein-protein interaction network, as essential genes should not have been identified in the first place. Moreover, THP-1 cells have gone through very few doublings before the sorting for phagocytosis was conducted. This implies that the time after infection may not have been sufficient for essential genes to drop-out. In this context, it is unclear how essential genes were filtered out before downstream analysis and confounds some of the statements in the manuscript.

We thank the referee for this comment and for the suggestion on how to potentially handle the term essentiality in our paper. We apologize for the misleading statement around essentiality in the previous version of our manuscript and now addressed the problem in detail as elucidated in the general point above. In the revised manuscript we now included a detailed description of how essentiality was assessed (**Suppl Fig 1**, Methods **page 21** and in the main text **pages 6-7**) and compared our dataset to DepMap (described on **page 6**, Methods **page 27** and illustrated in **Suppl Figure 5**).

6. The authors used latex beads for their research. It is possible that some of the hits are driven by the physicochemical properties of latex beads, which could be different from those of microbes, apoptotic cells, and malignant cells in terms of size, rigidity, and curvature; factors that are known to influence phagocytosis? Further validation would improve the generalizability and impact of the study.

We thank the reviewer for this very valid assessment. We think that the diversity of identified complexes and their rather high level of completeness is indicative that our screen could capture a large variety of genes that are involved during cargo uptake as well as on the way of cargo uptake and transport through the cell. We however fully agree that there will be certain genes differently involved based on substrate. The ideal approach would therefore be to do several FACS-based genome-wide screens like ours with a whole set of different pHrodo-coated substrates. This is however still a very time- and cost-intensive process. In this project alone we used over 700 million cells per replicate, and it took us five years and a very large sum of money. To expand to other kinds of substrates is beyond the scope of this study, chosen to be depth over breadth. Methods that are less accurate but allow for faster separation are more suitable to test multiple substrates. Exactly this aspect was employed in the study of Haney et al., which we addressed in detail above, and which provides a good understanding of genes that are specifically involved in the uptake of a panel of different substrates. However, we fully agree with the reviewer, that the use of a single substrate is an important limitation of the present study and addressed this candidly on **page 18** in the revised manuscript. For WASF2 knock-out lines, we validate our data in addition to phagocytosis assays with *Listeria monocytogenes*.

7. Validation of hits were only done in THP-1 cells. Therefore, the findings cannot be generalized to "the human phagocytosis molecular machinery," as indicated in the title of the manuscript. We encourage the authors to validate the screen hits in other human monocytic cell lines (both monocytes and macrophages) as well as in primary human monocytes and monocyte-derived macrophages to confirm that the hits are regulating phagocytosis broadly across different models.

The reviewer raised an important point. The current manuscript includes a large panel of validation cell lines generated mostly in THP-1. In **Figure 3a** alone we generated 74 different stable expressing knock-out cell lines and performed for all of them phagocytosis assays in biological duplicates and technical triplicates. Some validation cell lines were however also generated in U937 cells, another monocytic cell line.

Strategically we think to address this problem by using publicly available protein-protein interaction data that was generated in the completely unrelated kidney cell line 293T, to construct our network and validation approach we further think that those identified complexes should also be present in more related cell types. Nevertheless, the molecular machinery is likely to have been generally covered, as a cellular machinery enabling phagocytosis is likely to be common to all phagocytosis-competent human cells and relatively well conserved in evolution (by some accounts even fundamental for all eukaryotic cells). Our very deep genetic screening and the coverage of so many protein complexes previously associated with the process stands to testify that. As the reviewer suggests, other cell types are likely to have additional bells and whistles but not a different molecular machinery. We modestly ask to be allowed to maintain the claim on the human phagocytosis molecular machinery. If we were to put in relation every time a molecular process has only been shown in a relatively narrow number of biological settings, the cell biology and biochemistry textbooks would be mostly empty. We are not new to global studies of eukaryotic cell machinery (e.g. we characterized "all" yeast cellular machines) and think that in this case, the use of machinery and the process address does not imply that we claim the "full complement of genes involved in all possible cell types" but that the screen and bioinformatic approach offers an extraordinarily good coverage of the human phagocytosis molecular machinery.

8. The authors confirm the eIF5a hypusination pathway (doi: 10.1038/s41598-021-92332-7) as a positive regulator of phagocytosis. However, it is not clear mechanistically how it may be affecting phagocytosis. That is, by promoting phagosome-lysosome fusion (as suggested by figure 4h since there is no change in phagoEarly fraction) or by inducing target engulfment and internalization? In addition, it would be helpful to assess the mRNA targets of hypusinated versus unmodified eIF5a and whether they may be implicated in phagocytosis, including phagosome maturation, in order to improve the findings of the manuscript.

We fully agree with the reviewer. We are indeed very excited about the hypusine pathway and in fact, this screen started a multi-year project that will hopefully allow us to fully understand the mechanisms behind the hypusine axis. For this study, we generated a doxycycline-inducible DHPS knock-out cell model, that allowed us to prove with cDNA add-back experiments, that the phagocytosis defect is specific for hypusine and can indeed be rescued by reintroducing DHPS with a cDNA. Additionally, we showed defective phagocytosis when we inhibited DHPS pharmacologically and expressed an antibody against hypusine to

validate the absence of hypusine in these conditions. While the hypusine pathway was indeed recognized in the elegant study mentioned above, none of these important experiments have been performed. At the moment, for this study, we are not able to provide further mechanistic insights, but we are processing several large-scale datasets that we generated around the hypusine axis and hope to provide further mechanistic insights in the future. We also share the excitement of the reviewer in looking at RNA immunoprecipitation of hypusinated versus unmodified eIF5A, which was however beyond the scope of this study.

Other comments:

• Fig 1b. The Red and YG signals increase with intensity implying potential compensation issues. Have the stains been properly compensated? In addition, have there been any experiments looking at the stability as a function of time for pHrodo Red signal. For how long can pHrodo emit signal in the lysosome before it is degraded? The increased pHrodo signal seems to be increasing with number of internalized beads. Does this imply that certain sub-populations of cells are hyperactive in not only phagocytosing, but also processing the cargo? This could be assessed by comparing YG-beads (no pHrodo) and beads-pHrodo (no YG signal).

As the referee already suspected, the YG and pHrodo-Red signals both indeed increase with the number of total beads in the individual cell. Therefore, one would expect to see more total pHrodo signal in cells with more beads. We set the gates for our phagocytosis assays based on the pharmacological controls Cytochalasin D (inhibiting uptake) and Bafilomycin A1 (blocking acidification), we believe that this setup represents a functional way of setting the gates.

As suggested, we looked also at the stability of the pHrodo signal over time and added this control to the revised manuscript (**Suppl Fig 2b** and mentioned it in the text on **page 6**). For this, we did a standard phagocytosis assay with differentiated THP-1 WT cells, stopped it after 3 hours, and either measured the sample right away or measured it after 24 hours (storing the cells at 4 °C). As one can see in **Suppl Figure 2b**, pHrodo is very stable and there is no significant change in our phagocytosis assays, even after storing the cells for 24 hours at 4 °C.

• Fig 2. A bit difficult to follow the color scheme of the figure. It would be better to have a 3-color scheme for the log₂(FC) (with 0 being white) with colors designating complex different from the log₂(FC) colors.

We changed the color of the complexes to light brown, a color completely different from the color palette used to illustrate the log₂-fold changes (log₂(FC)). However, since the network is based on the 716 genes that are depleted in PhagoLate compared to PhagoNeg all the average log₂(FC) values are negative. We hope that this explains why we chose the current palette, which was optimized to best show the effects in the range of the data (-0.24 to -2.29 average log₂(FC)).

- **Fig 3a.** *It would be more informative to show effect size for significant genes rather than FDR or p-value. Also, correct the p-value scale in Figure 3a, as $p = 0.04$ appears twice, and p {less than or equal to} 0.05 already includes 0.04 , 0.02 , 0.01 and 0 .*

We revised **Figure 3a** as requested and are now showing fold change (FC) overlaid with the p-value illustrated as asterisks. The scale now represents the FC and the mentioned mistake in the labeling of the old scale got removed.

- **Fig. 3b.** *We would expect to see the strongest change in the phagoEarly fraction since Arp2/3 is critical for the phagocytic cup formation. The authors should comment on the fact that the observed results in their model do not recapitulate the expected effect of Arp2/3 ablation.*

As pointed out by the referee Arp2/3 is critical for phagocytic cup formation (Goley & Welch 2006). Therefore, one would expect that cells with defective Arp2 machinery have an overall decreased phagocytosis rate and therefore a strong increase in the PhagoNeg fraction. In **Figure 3a-b** we compare cells knocked out for *ARPC2* to control cells with our phagocytosis assay. As expected, when *ARPC2* is knocked out, significantly more cells fail to phagocytose and therefore end up in the PhagoNeg fraction. This is what one would expect and therefore our observed results recapitulate the expected effects for Arp2/3 ablation. The effect seen in PhagoLate is secondary to the uptake defect. If there is no uptake of the reporter, there will be also fewer cells with acidified reporter (PhagoLate). However, one would not expect direct effects of Arp2/3 on the acidification step of phagocytosis (PhagoEarly) and we also don't see that in our data.

- **Fig. 4f.** *There are several bands associated with DHPS. Do they correspond to different isoforms, PTMs or non-specific binding? As per the KO (lane 1 vs lane 6) it would appear that the top band corresponds to wt DHPS since it is larger in the Dox- compared to Dox+ condition. Quantification will be helpful.*

We updated **Figure 4f** in the revised manuscript and annotated the individual bands of the western blot. The top band represents the overexpressed *DHPS*-cDNAs. Overexpressed *DHPS* variants are a little bit larger due to a linker sequence on the C-terminus coming from the expression vector. This allows us to distinguish WT *DHPS* from overexpressed *DHPS*-cDNAs. The middle band represents the endogenous *DHPS* (*DHPS*^{WT}), according to size and its reduction in the inducible *DHPS* knock-out pools. Since we are working with knock-out pools that carry a sgRNA against *DHPS* and a Dox-inducible Cas9, we don't expect a full knock-out but rather a strong reduction of *DHPS* protein. This is what we can see if we compare lane 1 (+Dox) with lane 6 (no Dox). The last band we annotated as an unspecific band, we see it in all conditions, and knock-out of *DHPS* did not affect it. This is difficult to see in the exposure used in the manuscript, but can be better seen in a longer exposure that we are attaching below:

We quantified the western blot in the revised manuscript, added the ratio to **Figure 4f**, and described the used procedure in the methods on **page 25**.

- *It is not clear how THP-1 cells were differentiated exactly (how many days of 10nM PMA). There are many protocols for PMA-dependent differentiation of THP-1 cells into macrophages. Some protocols, particularly the ones employing only 10nM of PMA, are associated with a significant number of non-differentiated suspension cells, which will result de facto in a decrease in the fold gene coverage of each gRNA cassette for the CRISPR screen.*

We agree and have now incorporated the details about the PMA differentiation in the method section on **page 20**. Additionally, we generated a new illustration (**Suppl Fig 1**) that summarizes the timeline and conditions of the screen. Indeed, significant amounts of cells are lost during the PMA differentiation step (up to 50%). To collect enough genomic DNA after the FACS sorting step (to keep a constant cell number per column used to extract gDNA) we had to differentiate 750 million cells per replicate (coverage of ~9000x). This excess in cells allowed us to keep a gene coverage of each sgRNA after PMA differentiation far above the recommended 1000x (Doench 2018) and we could achieve reproducible screening replicates as seen in the PCA plots of **Suppl Fig 3b**.

- *The authors focus on depleted gRNA cassette reads from the phagoLate fraction only to determine positive regulators of phagocytosis. What about the enriched gRNA cassettes in phagoLate? What is the overlap between depleted in phagoLate and enriched in phagoNeg?*

We did not analyze for sgRNAs enriched in PhagoLate, as genes that when knocked-out lead to an increase in phagocytosis typically require a different approach of validation. However, we indeed analyzed sgRNAs enriched in PhagoNeg and are illustrating their intersection below and attached now the hitlist to **Suppl Table 1** in the revised manuscript. Out of 248 genes enriched in PhagoNeg vs Input, almost all (208) are contained in our PhagoLate vs PhagoNeg hitlist, which is a good indication that our chosen dataset contains most phagocytosis regulators identified in the screen.

- ***"Interestingly, the knock-out of VPS35, a subunit of the retromer complex, significantly reduced the PhagoNeg and PhagoEarly fraction, but not the PhagoLate fraction, confirming in a functional assessment of the different phagocytosis steps, that VPS35 regulates the uptake of materials to the phagosome and less the later part of phagocytosis." (pp10) The effect size for this hit is not indicated on figure 3a and the effects on phagocytosis for this hit have not been experimentally validated.***

We thank the reviewer for pointing out this interesting finding. We now changed **Figure 3a** to show in addition to the p-value also the direction of change of all assays. With this revised figure, one can now see that VPS35 knockout leads to a reduction of PhagoNeg fractions and an increase of PhagoEarly fractions, indicating that VPS35 operates at the interface between early and late endosome, compatible with the reported role of retromer in the literature (Seaman 2012). We also updated our manuscript to discuss this aspect on VPS35 on **page 10**.

- ***"Through the work presented here, it is possible to ascribe a biological role in phagocytosis to the CCC/Commander complex, likely to be related to the vesicular sorting function of the retromer complex (Figure 2, Figure 3a)." (pp10) Very little/no validation shown for this statement.***

We fully agree that this was a rather speculative statement and changed the wording of the statement now accordingly on **page 11**.

Point-by-point address to Referee #2

(*italics* are reviewer comments)

In this manuscript, entitled "A genome-wide CRISPR functional survey of the human phagocytosis molecular machinery", Essletzbichler et al. use a FACS reporter-based pooled genome-wide CRISPR knockout screen to dissect the positive genetic interactors of phagocytosis in a human myeloid cell line. The authors score 716 targets, mapping them elegantly to a protein-protein interaction network, which facilitates the contextualization of single hits in different functional protein clusters and serves as a landmark to guide a conservative and rational selection of hits for validation. The validation of 18 different targets belonging to the most represented/complete functional complexes with single KO experiments and, in few instances, with add back experiments, confirms overall a successful screening strategy. The large set of targets identified in the genetic screen belong mostly to well-known protein complexes regulating several aspects of phagocytosis, starting from engulfment, and spanning through trafficking and signaling of endosomes and lysosomes, as well as N-linked protein glycosylation. In some cases, in addition to previous findings, this work expanded the number of members of a specific pathway/protein complex identified as in the case of the magnesium transporter MAGT1, and specifically confirmed the emerging role of the eIF5A/hypusination pathway in the context of phagocytosis.

Overall, this work corroborates and expands recent studies on this subject and provides a good example for the visualization and rationalization of functional connections among hits of a genetic screen. The manuscript is written in a clear form and well-presented figures; the phagocytosis assay setup, the screen analysis and the validation experiments are robust enough to support a correlation between screen significance score and the functional role of the hit in phagocytosis.

Even though the threshold set to identify hits has a relative loose stringency, I can still appreciate the biological significance of the candidates to phagocytosis thanks to the very meaningful integration of the average sgRNA scores per gene to the protein interaction map.

We thank the reviewer for appreciating our manuscript, both at the experimental and written level. We addressed all the raised concerns in this revised manuscript, as described below.

Specific Comments:

-Can the authors describe how they defined gene essentiality in their screen? For clarity a step-by-step process of analysis should be described and included in methods, including which datasets were compared and the rationale for this choice.

It remains unclear how the authors define essentiality and why genes like Rab7A, appears both as hit in their screen (table S1) and in their essentiality list (table S2). Please clarify.

-On the same line of thought, how do the authors explain the discrepancy between their gene essentiality analysis and essentiality scores reported for the same cell line (THP-1) in the database DeepMap (<https://depmap.org/portal/>) for example? For instance, in the case of eIF5A and DHPS the hits do not appear in the essentiality list while they score as essential in DeepMap.

We thank the referee for this request and agree that our manuscript lacked details around estimating essentiality within our screen. We, therefore, addressed this issue in detail above as general point, added details to the methods and text (**Suppl Fig 1**, Methods **page 21** and in the main text **pages 6-7**), and compared our dataset to DepMap (described on **page 6**, Methods **page 27** and illustrated in **Suppl Figure 5**).

Furthermore, we looked up eIF5A, DHPS, and DOHH in DepMap and plotted the data below. The plots show the DepMap Chronos scores on the x-axis and the gene name in the title. The red line at -1, highlights the threshold for essential genes, and the orange-colored dot highlights the gene effect value for that gene for THP-1 (the cell line used for our screen). While it is true that the DepMap Chronos score for DHPS and eIF5A is essential on average, we don't see essentiality in DepMap for any of the three genes for THP-1, the cell line used for our screen. However, it is clear from this data, that growth effects in knock-out cells of the hypusine axis may play a role, which is why we generated our inducible DHPS knock-out model (**Figure 4f,h**).

-In Fig 1.b: (I) Can the authors explain or eventually show what was the pre-gating strategy? Were cell singlets pre-gated or was performed any sort of nuclear or viability staining? It is not reported in the methods section and I would recommend for it to be added.

We agree with the reviewer that this is an important point and now added information about the FACS pre-gating strategy to the methods section ("*Phagocytosis assay*") on **page 19**.

(II) Since the % of cells for each bin (phago-negative/early/late) are used to calculate statistics and infer significance, is the number of the pre-gated cells equal and normalized across samples? (e.g. using a down sampling algorithm).

The number of pre-gated cells is equal across all collected FACS samples in this study and was always 30'000 pre-gated recorded events. We added this important detail now to the methodology ("Phagocytosis assay") on **page 19** and thank the referee for spotting this.

(II) If only singlets were pre-gated, is the "striped" pattern in the "YG" channel actually showing an increasing, discrete number of beads binding to single cells?

We thank the referee for pointing out this additional information about the phagocytosis assay. Yes, the "striped" pattern in the YG channel indeed represents the individual beads as characterized nicely by the authors of the paper that described this phagocytosis assay in detail (Colas et al. 2014). We agree that this should be explained, and we added to the revised manuscript a description in the figure legend of **Figure 1b**.

-In Fig 1.c and any other figure reporting this type of analysis, I would advise strongly to plot data relative to each replicate (e.g. as single dots, superimposed) in addition to the error bar/confidence interval. For better interpretation of the data, the authors should specify whether the replicates included are independent biological experiments or technical replicates within the same assay.

We thank the referee for this suggestion and have incorporated your comments, by including in addition to the error bars also individual replicates as single superimposed dots. Additionally, we specified whether the replicates are technical or biological replicates in each experiment.

-In Fig 2 I believe neither the legend nor text specify what the very big light orange circles represent, unless I am missing it, this information should be included.

We agree that we did not describe the light orange ellipses that underlay the complexes with the highest completeness. We apologize. We added this missing information now in the legend of **Figure 2** and described them in the results section of the revised manuscript on **page 8**.

-In Fig 3a: While it is visually nice I have a few issues with this heatmap:

(I) each cell represents 3 technical replicates and a total of two biological replicates were performed. Correct? Please clarify this point in the legend and for Fig 3b.

Yes, this is correct, we thank the reviewer for pointing this out. As suggested, we added a better description of the biological and technical replicates used for this heatmap to the legend of

Figure 3a and **Figure 3b** and all other figure legends with similar data. We hope this takes care of the shortcoming.

(II) The heatmap shows a statistical difference but not the direction of the difference - so we must assume always that there is only one direction: for the KOs increased % of cells in the "phagoNegative" and "phagoEarly" and decreased in the "phagoLate" compared to control. Is this confirmed for all gene specific KOs pools since is not visible in the heatmap?

We thank the reviewer for highlighting this and agree that the previous heatmap did only include the statistical significance but not the direction of the difference. We now generated a new heatmap that shows in each cell in addition to the p-value, the average fold change (FC) of the three technical replicates (e.g. $[\%(\text{PhagoNeg})_{\text{ARPC2sg1}}] - [\text{mean } \%(\text{PhagoNeg})_{\text{Ctrl}}] / [\text{mean } \%(\text{PhagoNeg})_{\text{Ctrl}}]$). Blue color indicates a negative FC, white color no FC, and red color a positive FC. **Figure 3a** in the revised manuscript now contains this updated heatmap.

(III) The incongruence within replicates suggests that KO induced defect in phagocytosis is often of modest entity and therefore in general this assay would have benefited of a larger number of biological replicates which in turn would yield more robust and/or ponderate interpretation of the penetrance of the KO phenotypes. The authors should list in addition to the COPS, P4HA1 and PSMD13 also COG6, ATP6AP2, COMMD3 and any other target that did not reproduce at least in duplicate. If this is correct, the authors should rectify the sentence "In summary, of the genes that we assessed individually in phagocytosis assays, tracking three different stages, 83%..." to reflect either the correct percentage of genes that reproduced in duplicate experiments or a quantitative adjective.

We thank the referee for this valuable insight and certainly agree that yet more biological replicates would benefit the overall interpretation of the KO phenotypes. However, our goal was to trial our large phagocytosis dataset on a fair but still manageable scale. We believe that our approach of using two different sgRNAs for each validation target gene, followed by performing phagocytosis assays in two individual biological replicates consisting of three technical replicates each, should be a sufficient good proxy to test the quality of the dataset and the constructed network.

The referee points out that in addition to COPS, P4H1A1 and PSMD13 we should also list COG6, ATP6AP2, COMMD3, and any other target that did not reproduce at least in duplicates to the list of "non-validating" targets. We have incorporated this suggestion by adding two more genes to the "non-validating" targets list (KXD1 and COMMD3) and in addition commented or extended the previous interpretation of the effect of COG6, ATP6AP2, COMMD3, KXD1 in the text (**pages 9-12**). Additionally, the new **Figure 3a** now includes fold change of the phagocytosis fractions and allows to distinguish if variation in significance appeared due to varying directions of signals or due to very low levels of effect overall in that fraction.

-In Fig 3d: For clarity, the authors should specify in the methods section more details about the image quantification process of the bacterial infection, including number of fields used per experiment, method used for automated counting and whether any sort of normalization was performed (e.g. number of bacteria per cell).

We agree with the referee and have reflected on this comment by incorporating these details now into the methods sections on **page 24**.

-I recommend that the authors describe the content of the supplementary files, including a legend describing each column header.

We thank the reviewer for this comment and now added a legend describing each column to the supplementary files.

-I cannot help but wonder whether the authors performed the analysis to identify enriched sgRNA in the "phagoLate" fraction as compared to "phagoNegative" or "input" fractions as would possibly provide further insights about negative regulators of phagocytosis as reported in several previous studies, also referenced in this manuscript. Unless this is a reserved subject for a future study, I would suggest including it in this manuscript if available.

We thank the referee for these suggestions. For this paper, we decided to focus on the comparison that gives us the largest list of positive phagocytic regulators and then use our network integration approach based on public protein-protein interaction data to group them into complexes. We might indeed follow up in the future with a similar approach focusing just on negative regulators.

We have however included now the hit lists for genes depleted in PhagoLate compared to Input and enriched genes In PhagoNeg vs Input to **Suppl Table 1**.

Minor:

-please describe in more detail the treatment of cells with PMA, number of days of treatment is lacking.

We added the details to the methods section (**page 20**) and generated **Suppl Figure 1** which gives now a detailed overview of the used protocol, cell numbers, and conditions.

- Across the manuscript "Welch" instead of Welsh.***
- ***"accumulation of toxic disease proteins" =toxic accumulation of proteins***
- ***"expluded" = excluded***
- ***"amendable" = amenable***

We are thankful for pointing out these mistakes and corrected them across the manuscript.

- ***"its novel link to phagocytosis" = while some members of the OST complex are newly associated to phagocytosis, the OST complex has been previously associated with phagocytosis (e.g. in Haney et al.), hence the word novel should be avoided in this context, or the sentence rephrased.***

We have removed the part of the sentence classifying it as novel.

- ***"abolishe" =abolish***

We now corrected the mistake across the manuscript.

Point-by-point address to Referee #3

(*italics* are reviewer comments)

The manuscript by Essletzbichler et al. reports:

I. the results of a CRISPR-CAS9 screen on the phagocytic activity of THP-1 cells designed to distinguish genes that are required for the internalization step (whose depletion leads to a phago-negative phenotype) from those required for phagosome maturation/fusion with lysosomes (whose depletion leads to a phago-early phenotype);

II. an integrated analysis of the genetic hits that emerged from the screen with the existing protein-protein interaction data involving those hits, which led to the definition of a comprehensive network of 490 genes controlling the phagocytosis process;

III. the validation of the top hit from the screen, the hypusine axis.

Overall, the data are of very high quality and fully support the conclusions. However, they also raise some concerns that should be addressed.

We thank the referee for the overall very positive evaluation of our work and have addressed the comments point-by-point here below.

1. Several of the hits described in the manuscript were identified in previous genome-wide screens on the phagocytosing ability of macrophages. Hence, a part of the data confirms published data, so the authors should reduce the description of confirmatory data to expand and highlight what their study adds to the pre-existing published data.

We thank the reviewer for raising this point and agree that while we worked on this manuscript, multiple similar studies have been released. We made sure to mention all these previous studies in our manuscript and compared our data to them. As some parts of the OST complex have been indeed identified before in a phagocytosis screen, we removed the classification “novel” in its context in the text. Compared to previous studies we chose a different data integration approach that uses the protein-protein interactions of identified genetic hits. This allowed us to work with a more complete list of phagocytic regulators. The advantage of this approach can be seen for example on the Arp2/3 complex which is very well known to be responsible for phagocytic cup formation and can almost be seen as a positive control for the screening approach. While we identified 7/7 subunits of the Arp2/3 complex the “big beads” dataset of Haney et al. could identify 4/7 subunits. We, therefore, conclude that the same must be true for other identified complexes in the datasets and are convinced that our study is a good extension and complementation to the current literature.

2. One innovative aspect of the screen concerns the discrimination of the phago-negative from the phago-early phenotype, and this aspect should be exploited more. In fact, the authors do not exploit the power of their assay. They could extend the analysis of the hits inducing the phago-early phenotype. Considering that the distinction between early and late phagosome phenotypes is based on the pH of bead-containing organelles, they should assess whether the early-phenotype

is due to phagosomes that cannot fuse with properly acidified lysosomes or to phagolysosomes with a less acidic pH.

We fully agree with the referee's comments. However, the reason for not exploiting this part of the assay was a technical one. The total size of the PhagoEarly population is rather small, which leads to low amounts of total extractable gDNA and therefore does not allow performing comparable PCR and NGS prep to the other samples. We, therefore, decided to focus on the high-quality data derived from comparing PhagoLate versus PhagoNeg and now added all additional derived comparisons to **Suppl Table 1**. To collect more cells in the PhagoEarly state one would need to perform the screen at an even larger scale (we worked with 750 M cells per replicate), which then requires several days of FACS sorting per replicate. However, we agree that this could be a very interesting screen for the future when these technical limitations can be overcome.

3. In the workflow followed in the manuscript, the PMA treatment (for a time that is not specified) that induces the macrophage differentiation of THP-1 cells has been performed 4 days after the infection of THP-1 cells with lentiCRISPRv2 carrying Cas9 and sgRNA. Thus, one important control that should be performed is to check whether some of the genes that were hits in the phagocytosis assay were indeed affecting the differentiation process of THP-1 into macrophages. This control could be performed by analyzing the markers commonly monitored during this differentiation process (CD86, CD11, CD14).

We thank the reviewer for raising this important point and apologize for not adding these details to the originally submitted manuscript. We added our PMA differentiation protocol to the methods part (**page 20**) and in addition, generated a flowchart (**Suppl Fig 1**) illustrating the workflow of the genetic screen.

For our analysis, we are comparing sgRNA abundance in PhagoLate versus PhagoNeg or versus Input. Cells used as input for the FACS sorting are already differentiated. Since we are washing cells extensively after differentiation and before sorting, we expect genes affecting differentiation itself to not be included in the input anymore, which makes it unlikely that they score as genetic hits in our phagocytosis regulators dataset. We looked for CD86, CD11 (ITGAM) and CD14 in our datasets and indeed didn't find them as hits.

Literature

Colas C, Menezes S, Gutiérrez-Martínez E, Péan CB, Dionne MS, Guermonprez P. 2014. An improved flow cytometry assay to monitor phagosome acidification. *J. Immunol. Methods.* 412:1–13

DepMap B. 2022. DepMap 22Q2 Public. *Figshare*

Doench JG. 2018. Am I ready for CRISPR? A user's guide to genetic screens. *Nat. Rev. Genet.* 19(2):67–80

Goley ED, Welch MD. 2006. The ARP2/3 complex: an actin nucleator comes of age. *Nat. Rev. Mol. Cell Biol.* 7(10):713–26

Haney MS, Bohlen CJ, Morgens DW, Ousey JA, Barkal AA, et al. 2018. Identification of phagocytosis regulators using magnetic genome-wide CRISPR screens. *Nat. Genet.* 50(12):1716–27

Seaman MNJ. 2012. The retromer complex - endosomal protein recycling and beyond. *J. Cell Sci.* 125(Pt 20):4693–4702

January 16, 2023

RE: Life Science Alliance Manuscript #LSA-2022-01715R

Prof. Giulio Superti-Furga
CeMM Research Center for Molecular Medicine
CeMM Research Center for Molecular Medicine of the Austrian Academy of Sciences
Lazarettgasse 14
Vienna 1090
Austria

Dear Dr. Superti-Furga,

Thank you for submitting your revised manuscript entitled "A genome-wide CRISPR functional survey of the human phagocytosis molecular machinery". We would be happy to publish your paper in Life Science Alliance pending final revisions necessary to meet our formatting guidelines.

A. FINAL FILES:

B. MANUSCRIPT ORGANIZATION AND FORMATTING:

****The license to publish form must be signed before your manuscript can be sent to production. A link to the electronic license to**

publish form will be sent to the corresponding author only. Please take a moment to check your funder requirements.**

Sincerely,

Reviewer #2 (Comments to the Authors (Required)):

The authors have addressed satisfactorily all my previous comments. Figures and legends have been updated and the revised text throughout the manuscript improved significantly the clarity and interpretation of the findings. I have no further comments at this time.

January 19, 2023

RE: Life Science Alliance Manuscript #LSA-2022-01715RR

Prof. Giulio Superti-Furga
CeMM Research Center for Molecular Medicine
CeMM Research Center for Molecular Medicine of the Austrian Academy of Sciences
Lazarettgasse 14
Vienna 1090
Austria

Dear Dr. Superti-Furga,

Thank you for submitting your Research Article entitled "A genome-wide CRISPR functional survey of the human phagocytosis molecular machinery". It is a pleasure to let you know that your manuscript is now accepted for publication in Life Science Alliance. Congratulations on this interesting work.

DISTRIBUTION OF MATERIALS:

Again, congratulations on a very nice paper. I hope you found the review process to be constructive and are pleased with how the manuscript was handled editorially. We look forward to future exciting submissions from your lab.

Sincerely,
